# RAVR: Reference-Answer-guided Variational Reasoning for Large Language Models

## Abstract

Reinforcement learning (RL) can refine the reasoning abilities of large language models (LLMs), but critically depends on a key prerequisite: the LLM can already generate high-utility reasoning paths with reasonable probability. For tasks beyond the LLM's current competence, such reasoning path can be hard to sample, and learning risks reinforcing familiar but suboptimal reasoning. We are motivated by the insight from cognitive science that *Why is this the answer* is often an easier question than *What is the answer*, as it avoids the heavy cognitive load of open-ended exploration, opting instead for explanatory reconstruction—systematically retracing the reasoning that links a question to its answer. We show that LLMs can similarly leverage answers to derive high-quality reasoning paths. We formalize this phenomenon and prove that conditioning on answer provably increases the expected utility of sampled reasoning paths, thereby transforming intractable problems into learnable ones. Building on this insight, we introduce **RAVR** (Reference-Answer-guided Variational Reasoning), an end-to-end framework that uses answer-conditioned reasoning as a variational surrogate for question-only reasoning. Experiments in both general and math domains demonstrate consistent improvements over strong baselines. We further analyze the reasoning behavior and find that **RAVR** reduces hesitation, strengthens conclusion consolidation, and promotes problem-specific strategies in reasoning.

## 1 Introduction

Large language models (LLMs) can solve increasingly complex problems when guided by reinforcement learning (RL) (Zhang et al., 2025a). In this realm, a trajectory is a completion consisting of a reasoning path followed by a final response (Jaech et al., 2024; Guo et al., 2025; Yang et al., 2025; Agarwal et al., 2025). The objective is straightforward: to sample completions from the model's current distribution and then shift probability mass toward those with higher advantage, such as ones that produce a correct answer. This process is more like redistributing probabilities among sampleable completions, rather than generating entirely new ones (Yue et al., 2025). This indicates a critical prerequisite for effective optimization—*the model must already be able to sample useful completions with non-negligible probability*. In recently popular relative-advantage approaches such as GRPO (Shao et al., 2024), this prerequisite becomes even stricter, since the advantage of each completion is defined relative to others—meaning that even weak completions can be reinforced as long as they are better than the rest. Unfortunately, tasks beyond the model's competence or outside its preferences make high-utility completions difficult to obtain (Zhang et al., 2025b; Li et al., 2025). As a result, training collapses into reinforcing a narrow set of familiar but suboptimal completions, while promising ones remain unexplored.

To address this issue, we advance a simple thesis: utilizing the reference answer can help derive good reasoning paths. While the reference answer is available in the training data, current methods use it only to compute reward; we argue that its potential can be more fully exploited. In cognitive science, *Why is this the answer* is often an easier question than *what is the answer* because it relieves the learner from the high cognitive load of *open-ended exploration* based solely on the question. Instead, it allows the learner to concentrate on *explanatory reconstruction*—tracing the logic that connects the question to the reference answer (Chi et al., 1989). For example, access to the answer can help learners detect errors to keep exploration, avoid overthinking when they are correct, and even engage in backward reasoning to obtain the preconditions leading to the answer.

LLM is possibly capable of simulating this human behavior. We validate this intuition with a motivation experiment as shown in Figure 1: for many hard questions where the LLM fails to sample a valid reasoning to solve the problem correctly after multiple attempts, providing the answer enables it to generate rational reasoning.

Building on this insight, we propose using the reference answer as a guide to better explore high-quality reasoning. We formalize this effect and provide a proof in Section 2.2. Conditioning on the answer increases the likelihood of sampling sound reasoning and reduces the likelihood of sampling flawed reasoning, thereby improving the overall expected quality of reasoning over the sampling distribution.

While this is conceptually clean, this reasoning generation pipeline does not align with the real need at inference stage—the LLM can only observe the question at inference and needs to generate the reasoning instantly in the thinking tags. We address this issue from the perspective of variational inference (Gershman & Goodman, 2014), viewing the *derived reasoning* conditioned on the reference answer as a

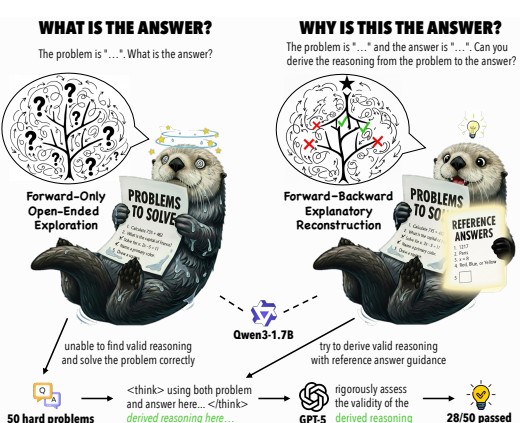

Figure 1: We sampled 50 hard questions from CrossThink-QA (Akter et al., 2025) on which Qwen3-1.7B, with thinking mode enabled, failed to obtain the correct answer across 8 attempts. For each question, we provided the answer in the prompt and asked the LLM to derive the reasoning. A GPT-5 judge, using strict criteria, evaluated whether the derived reasoning was logically coherent without indicating access to answer. In over 50% of the unlearnable questions, the LLM was able to produce correct reasoning. See Appendix A.2 for detailed prompts and cases.

variational surrogate for the reasoning inside the thinking tags when question-only. We optimize the evidence lower bound (Kingma & Welling, 2013) by maximizing expected reasoning utility over the answer-conditioned distribution while reducing the discrepancy between the answer-conditioned reasoning distribution (hereafter referred to as *Posterior*) and the question-only reasoning distribution (hereafter referred to as *prior*) via Kullback–Leibler (KL) divergence (derivation in Section 2.3). To further stabilize training, we introduce the following designs. First, the language style can be different in the two distributions. The *derived reasoning* generated after the thinking tags can be concise and direct, but the internal reasoning under prior is usually exploratory and reflective. To bridge this gap, we leverage a role-play style prompt (Shanahan et al., 2023), asking the LLM to produce a first-person think-aloud monologue as if solving the problem from scratch. The monologue is allowed to include any classical reasoning behavior such as reflection and backtracking. This setup encourages the *derived reasoning* to mirror the behavior expected inside the thinking tags. Second, we introduce a utility baseline for estimating reasoning utility under the posterior, which is the expected reasoning utility under the prior. This provides a more informative reward signal by measuring how much the reasoning from the posterior improves over that from the prior. Third, we reweight samples in the KL estimation using the utility reward, with the goal of aligning the prior to a distribution whose expected utility is as large as possible. With the proposed variational objective and the designed strategies, we can finally end-to-end train a single LLM that learns to reason effectively from scratch at inference.

We name this framework as **RAVR**, for **R**eference-**A**nswer-guided **V**ariational **R**easoning. We evaluated **RAVR** under two training settings—math data and general data. Results on standard benchmarks for both general and math reasoning show that **RAVR** substantially enhances reasoning capabilities and outperforms state-of-the-art methods. For example, when we use **RAVR** to train Qwen3-1.7B on the CrossThink-QA (Akter et al., 2025) dataset, it achieves a GPQA-Diamond score of 40.91, outperforming DAPO (Yu et al., 2025a) by 5.56 points. Additionally, we analyze the reasoning behavior of the LLM trained with **RAVR** and find that **RAVR** can reduce hesitation, strengthen conclusion consolidation, and promote problem-specific strategies in reasoning.

Overall, this paper makes the following contributions: (1) We formalize the intuition that reference answers can guide reasoning. We prove that conditioning on the answer provably amplifies the probability of high-utility reasoning paths; (2) We propose **RAVR**, the first end-to-end framework that operationalizes this insight and also the first to leverage the reasoning ability of the LLM the use

the reference answer. **RAVR** alleviates the exploration difficulty in RL for LLM; (3) We demonstrate the effectiveness of **RAVR** through extensive experiments on both general and math domains and the in-depth analysis of the reasoning behavior of the LLM trained with **RAVR**. We share our code to facilitate future research.

## 2   **RAVR**: REFERENCE-ANSWER GUIDED VARIATIONAL REASONING

In this section, we introduce the background of RL for LLM, explain the motivation of **RAVR** and then describe the proposed objective and strategies to realise the motivation.

### 2.1   PRELIMINARIES AND THE SAMPLING CHALLENGE

In this work, we consider training an LLM, denoted by $\pi_\theta$, to be a Large Reasoning Model (LRM). Given a sample $(x, y^*)$ from a dataset $\mathcal{D}$, the model is supposed to produce an intermediate *reasoning path* $z$ before giving a final answer $y$ for each problem $x$. This can be factorized as:

$$\pi_\theta(z, y \mid x) = \pi_\theta(z \mid x)\, \pi_\theta(y \mid x, z). \tag{1}$$

In general setting, a reward function is designed, which provides a scalar reward $R(y) \in \mathbb{R}$ to judge the consistency of the reference answer $y^*$ and the generated answer $y$. The standard objective is to maximize the expected reward:

$$\mathcal{J}(\theta) = \mathbb{E}_{z \sim \pi_\theta(\cdot \mid x)}\, \mathbb{E}_{y \sim \pi_\theta(\cdot \mid x, z)}\big[R(y)\big]. \tag{2}$$

To align with the comparative nature of reward models, rencent popular RL algorithms for LLM usually normalize the reward of each completion in a group-relative manner to obtain the advantage of each completion. Taking GRPO (Shao et al., 2024) as an example, it first generate multiple completions for a problem $x$, i.e., $z$ and $y$, and calculates the advantage of $i$-th completion as $A_i = \frac{R_i - \text{mean}(R)}{\text{std}(R)}$, where $\text{mean}$ and $\text{std}$ represent the average and standard deviation of the rewards. *This indicates a critical prerequisite for effective optimization—the model must already be able to sample good completions with non-negligible probability*, because even weak completions can be reinforced as long as they are better than the rest and obtain a positive advantage. When few good completions are sampled, training risks collapsing into reinforcing familiar but suboptimal completions. Recent work also proposes to normalize the reward at batch-level (Hu et al., 2025). This alleviates the challenge at the problem level, yet still faces the challenge at dataset level. If $R(\cdot)$ is defined as binarized correctness, which is common in current applications, this challenge can be more severe when no correct completion is generated since all the advantages become zero, making no optimzation signal.

### 2.2   MOTIVATION: CONDITIONING ON REFERENCE ANSWER AMPLIFIES GOOD REASONING

The reasoning path $z$ dominates the final quality of the entire completion since it causally influences the generation of the final answer $y$. Therefore, we propose to mitigate the sampling challenge of high-utility reasoning paths to alleviate the sampling challenge of the overall completion. In this section, we formally define the utility of a reasoning path, and prove our motivation that conditioning on reference answer can amplify the sampling probability of high-utility reasoning paths.

If $R(\cdot)$ denotes binarized correctness, optimizing reasoning path $z$ aims to maximize the expectation it yields the reference answer. Hence, a natural alternative is to take the LLM's likelihood of the reference-answer as reward on $z$ and the LLM can be optimized by maximizing the expected reward:

$$\mathcal{J}_{\text{prob}}(\theta) = \mathbb{E}_{z \sim \pi_\theta(\cdot \mid x)}\big[R_{\text{prob}}(z)\big], \quad \text{where } R_{\text{prob}}(z) = \pi_\theta\big(y^* \mid x, z\big). \tag{3}$$

This objective is not fully match the original setting, for example, the reference answer $y^*$ is usually a single word or phrase such as an option "A" of a multiple-choice question, but the prediction $y$ can be a paragraph that includes steps to reach the answer. Yet, recent works have shown its acceptable effectiveness (Yu et al., 2025b; Zhou et al., 2025b), especially in open-ended domain (Xu et al., 2025; Wang et al., 2025b). Therefore, $R_{\text{prob}}$ provides a reliable operational measure of the utility of a reasoning path $z$. Let's formally define the utility score as:

$$s(z) := \pi_\theta(y^* \mid x, z) \in [0, 1]. \tag{4}$$

Then we can define the ability of the LLM as the expected utility over its reasoning distribution:

$$\mu \ := \ \mathbb{E}_{z \sim \pi_\theta(\cdot|x)}[s(z)] \tag{5}$$

Note that this equals to $\pi_\theta(y^* \mid x)$ because of the law of total probability. We can define the $\tau$-*good* set as $\mathcal{Z}_\tau = \{z : s(z) \geq \tau\}$, where $\tau \geq \mu$ emphasizes above-average reasoning paths.

The current challenge is to more efficiently sample reasoning paths with higher $s(z)$. We can prove that observing $y^*$ can increase the sampling probability of reasoning paths with higher $s(z)$. To begin with, for a specific reasoning path $z$, by the law of total probability and Bayes' rule, we have

$$\mathbb{E}_{c \sim \pi_\theta(\cdot|x,y^*)} \pi_\theta(z|x,y^*,c) = \pi_\theta(z \mid x,y^*) = \frac{\pi_\theta(y^* \mid x,z)\,\pi_\theta(z \mid x)}{\pi_\theta(y^* \mid x)} = \frac{s(z)}{\mu}\,\pi_\theta(z \mid x), \tag{6}$$

where the first expression describes that the LLM thinks through the problem and reference answer jointly before producing the reasoning. It shows that *observing $y^*$ induces a size-biased reweighting of $\pi_\theta(z \mid x)$ by $s(z)$: high-$s(z)$ paths gain probability mass, low-$s(z)$ paths lose it.*

For any subset of reasoning paths $Z$, the probability of sampling one reasoning path belonging to $Z$ can be formulated as follows

$$\Pr(Z \mid x, y^*) = \sum_{z \in Z} \frac{\pi_\theta(y^* \mid x, z)\,\pi_\theta(z \mid x)}{\pi_\theta(y^* \mid x)} = \Pr(Z \mid x) \cdot \frac{\mathbb{E}[s(z) \mid z \in Z,\, x]}{\mu}. \tag{7}$$

See Appendix A.4 for detailed derivation. Consequently, we have

$$\Pr(Z \mid x, y^*) \ \geq \ \Pr(Z \mid x) \quad \Longleftrightarrow \quad \mathbb{E}[s(z) \mid z \in Z,\, x] \ \geq \ \mu. \tag{8}$$

For $\tau$-*good* set $\mathcal{Z}_\tau = \{z : s(z) \geq \tau\}$, we have

$$\frac{\Pr(\mathcal{Z}_\tau \mid x, y^*)}{\Pr(\mathcal{Z}_\tau \mid x)} \ \geq \ \frac{\tau}{\mu}, \quad \text{with strict increase if } \tau > \mu. \tag{9}$$

Therefore, conditioning on reference answer $y^*$ amplifies above-average reasoning paths. Beyond sets, the posterior raises the expected utility over the distribution, because given equation 6 and $\mathbb{E}\left[s(z)^2\right] = \mathbb{E}\left[s(z)\right]^2 + \mathrm{Var}(s(z))$, we can derive

$$\mathbb{E}_{z \sim \pi_\theta(\cdot|x,y^*)}[s(z)] = \frac{1}{\mu}\,\mathbb{E}_{z \sim \pi_\theta(\cdot|x)}[s(z)^2] = \mu + \frac{\mathrm{Var}(s(z))}{\mu} \ \geq \ \mu. \tag{10}$$

In summary, approximate $\pi_\theta(z \mid x, y^*)$ acts as a principled target for exploring high-utility reasoning paths and thus it can be properly used to help the learning of $\pi_\theta(z \mid x)$.

## 2.3 REFERENCE-ANSWER-CONDITIONED VARIATIONAL OPTIMIZATION OBJECTIVE

Building on the above insight, we propose a novel variational objective, introducing an amortized posterior $\pi_\theta(z \mid x, y^*)$ to aid the learning of the prior $\pi_\theta(z \mid x)$. We start from log-transformation of the raw objective, which maximizes the utility score over the reasoning distribution:

$$\log \mathcal{J}(\theta) = \log \mathbb{E}_{z \sim \pi_\theta(z|x)}\left[\pi_\theta(y^* \mid x, z)\right]. \tag{11}$$

By introducing the amortized posterior $\pi_\theta(z \mid x, y^*)$ and applying Jensen's inequality, we can derive the Evidence Lower Bound (ELBO) of the objective as follows:

$$\log \mathcal{J}(\theta) \geq \mathbb{E}_{z \sim \pi_\theta(z|x,y^*)}\left[\log \pi_\theta(y^* \mid x, z)\right] - \mathbb{D}_{\mathrm{KL}}\left[\pi_\theta(z|x,y^*) \| \pi_\theta(z|x)\right]. \tag{12}$$

See Appendix A.5 for the derivation. The second term in equation 12 is the Kullback-Leibler (KL) divergence between the amortized posterior and the prior and we estimate it via the approximator introduced by Schulman (2020). The first term encourage reasoning paths that make $y^*$ more likely. The KL term pulls the prior toward the posterior and also regularizes the posterior to prevent it from collapsing onto out-of-distribution reasoning path for the prior.

To further stabilize training, we innovatively introduce a utility baseline for estimating reasoning utility under the answer-conditioned posterior: the expected utility under the question-only prior.

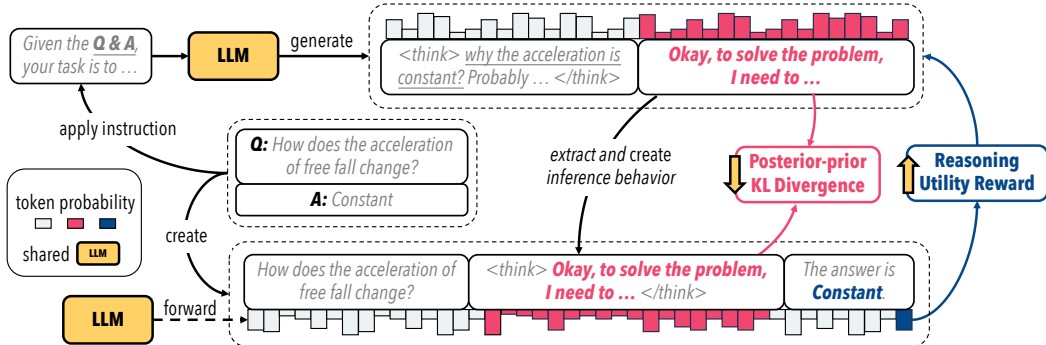

Figure 2: The framework of **RAVR**. According to Section 2.2, seeing the reference answer can amplify the sampling probability of good reasoning paths. Hence, we use this answer-conditioned posterior to help the learning of the question-only prior. The LLM is instructed to derive reasoning path from the question to the answer with thinking mode enabled. **RAVR** regard the reference-answer probability as the reward for the generated reasoning and maximize it to enhance the ability of the LLM to think *why is this the answer*. Meanwhile, **RAVR** minimize the KL divergence between the posterior and the question-only prior to help the model better think *what is the answer* and in turn, the prior also regularizes the behavior of the posterior. See Section 2.3 for details.

This provides a more informative reward signal by measuring how much the reasoning under the posterior improves over that under the question-only prior.

$$R_{\text{impr}}(z) = \max(0, \ \log \pi_\theta(y^*|x,z) - \mathbb{E}_{z' \sim \pi_\theta(\cdot|x)} \log \pi_\theta(y^*|x,z')). \tag{13}$$

We clip the minimum reward to zero to stabilize training. In practice, we replace the likelihood with length-normalized sequence likelihood. This avoids the issue where longer reference answers receive smaller rewards, which would make reference answer of different lengths incomparable. Moreover, we identify and address a subtle but significant bias in how existing probability-reward-based methods (Yu et al., 2025b; Zhou et al., 2025b) estimate answer likelihood. These methods compute $\log \pi_\theta(y^* \mid x, z)$ by directly appending the reference answer after the reasoning part, which fails to control for the model's internal state transition to answer generation. Our solution is to standardize this transition by inserting a cue phrase, estimating the probability as $\log \pi_\theta(y^* \mid x, z, \text{"The answer is "})$ within the structured sequence *"The answer is $y^*$"*. This answer-prefix not only aligns with common linguistic patterns but also establishes a consistent benchmark for probability estimation across all reasoning paths, thereby guaranteeing the validity of the group-relative normalization that follows.

Moreover, once the model has mastered high-quality reasoning for a given problem, it no longer needs external guidance and should learn only from paths with higher utility than its own. Accordingly, we apply reward-based weighting to samples when estimating the KL.

$$\tilde{\mathbb{D}}_{\text{KL}}\left[\pi_\theta(z|x,y^*)\|\pi_\theta(z|x)\right] = R_{\text{impr}}(z) \cdot \mathbb{D}_{\text{KL}}\left[\pi_\theta(z|x,y^*)\|\pi_\theta(z|x)\right] \tag{14}$$

Therefore, the final variational objective is given as:

$$\mathcal{J}_{\textbf{RAVR}}(\theta) = \mathbb{E}_{z \sim \pi_\theta(z|x,y^*)}\left[R_{\text{impr}}(z)\right] - \tilde{\mathbb{D}}_{\text{KL}}\left[\pi_\theta(z|x,y^*)\|\pi_\theta(z|x)\right]. \tag{15}$$

We utilize GRPO (Shao et al., 2024) to optimize the first term. In practice, we jointly optimize this objective and Equation 11. To realize the two distribution $\pi(\cdot|x)$ and $\pi(\cdot|x,y^*)$ within one LLM, we adjust the user prompt. Details of the template are as follows.

**Prompt Template for Question-only**

<|im_start|>system

A conversation between user and assistant. The user asks a question, and the assistant solves it. The assistant first thinks about the reasoning process in the mind and then provides the user with the answer. The reasoning process is enclosed within <think></think> tags, i.e., <think>This is my reasoning. </think>This is my answer.<|im_end|>
<|im_start|>user
{{question}}<|im_end|>
<|im_start|>assistant

---

**Prompt Template for Conditioning on both Question and Reference Answer**

\# same system prompt.
<|im_start|>user
Given the following question and its reference answer, your task is to produce a step-by-step explanation that logically leads to the reference answer, written in the style of a first-person think-aloud monologue. You are encouraged to draw on the reference answer for internal guidance to help structure and support your reasoning, but the final monologue must read as a genuine, first-encounter, real-time discovery, without mentioning or implying any prior access to the reference answer.
Question: {{question}}
Reference Answer: {{ground_truth}}
OUTPUT REQUIREMENTS:
1. Output ONLY the first-person, think-aloud monologue. Do not include any preface, summary, or restatement of these instructions.
2. Maintain the tone of a focused individual thinking to themself. Avoid meta-commentary like "for the first time," and any phrasing that reveals simulation.
3. Do not mention, imply, or hint at prior access to the Reference Answer in the monologue. Avoid phrases like "according to the answer..." or "to get to that answer...", and any euphemism that signals foreknowledge.
4. Do not merely restate the final answer in the monologue; articulate the reasoning pathway with sufficient intermediate steps, rationale, decision points, verification, and any necessary error-correction or backtracking.
<|im_end|>
<|im_start|>assistant

## 3 EXPERIMENTS

In this section, we evaluate the effectiveness of **RAVR** on both general-domain and math domain. Additionally, we also analyze the learning dynamics of the method and efficacy of different components and the reasoning behavior of the model optimized by **RAVR**.

Table 1: Experiments on Qwen3-1.7B. General task represents average of GPQA-D and MMLU-Pro, and Math reasoning represents average of AIME24, AIME25, AMC23 and Minerva. The metric is the average@k accuracy, with the subscript indicating the corresponding average token length.

| Training Set | Model | GPQA-D Avg@4 | MMLU-Pro Avg@4 | AIME 24 Avg@16 | AIME 25 Avg@16 | AMC 23 Avg@16 | Minerva Avg@16 | General Task | Math Reasoning | Average |
|---|---|---|---|---|---|---|---|---|---|---|
| | Qwen3-1.7B | $21.46_{(6948)}$ | $53.48_{(3499)}$ | $20.00_{(7805)}$ | $23.30_{(7654)}$ | $55.00_{(5896)}$ | $50.00_{(5233)}$ | 37.47 | 37.08 | 37.21 |
| | | **In-domain** | | **Out-of-domain** | | | | | | |
| **CrossThink-QA** | + GRPO | $34.97_{(4581)}$ | $55.18_{(2587)}$ | $25.63_{(7273)}$ | $23.54_{(7283)}$ | $\mathbf{64.53}_{(4799)}$ | $56.92_{(3231)}$ | 45.08 | 42.65 | 43.46 |
| | + DAPO | $35.35_{(5991)}$ | $54.75_{(5129)}$ | $23.33_{(7556)}$ | $21.88_{(7495)}$ | $62.50_{(5502)}$ | $56.80_{(4129)}$ | 45.05 | 41.13 | 42.44 |
| | + VeriFree | $28.60_{(6092)}$ | $53.32_{(2672)}$ | $23.54_{(7532)}$ | $20.83_{(7386)}$ | $62.19_{(5044)}$ | $55.79_{(3529)}$ | 40.96 | 40.59 | 40.71 |
| | + RLPR | $30.93_{(5357)}$ | $53.82_{(2541)}$ | $24.83_{(7275)}$ | $22.92_{(7185)}$ | $62.81_{(4933)}$ | $56.34_{(3131)}$ | 42.38 | 41.73 | 41.94 |
| | + **RAVR** (Ours) | $\mathbf{40.91}_{(4177)}$ | $\mathbf{55.88}_{(2430)}$ | $\mathbf{27.92}_{(7236)}$ | $\mathbf{23.75}_{(7281)}$ | $62.03_{(5073)}$ | $\mathbf{58.00}_{(3320)}$ | **48.39** | **42.92** | **44.75** |
| | | **Out-of-domain** | | **In-domain** | | | | | | |
| **DeepMath** | + GRPO | $33.21_{(3932)}$ | $54.55_{(1934)}$ | $26.67_{(6935)}$ | $21.46_{(6906)}$ | $67.03_{(4602)}$ | $56.99_{(2766)}$ | 43.88 | 43.04 | 43.32 |
| | + DAPO | $\mathbf{34.84}_{(5242)}$ | $55.25_{(2684)}$ | $26.67_{(7263)}$ | $23.33_{(7070)}$ | $67.50_{(5043)}$ | $57.97_{(3633)}$ | 45.05 | 43.87 | 44.26 |
| | + VeriFree | $30.30_{(1256)}$ | $52.90_{(762)}$ | $26.88_{(2901)}$ | $21.88_{(2443)}$ | $62.34_{(1490)}$ | $55.01_{(713)}$ | 41.60 | 41.53 | 41.55 |
| | + RLPR | $31.31_{(5464)}$ | $54.27_{(2558)}$ | $26.04_{(7318)}$ | $22.50_{(7224)}$ | $62.19_{(4919)}$ | $55.51_{(3168)}$ | 42.79 | 41.56 | 41.97 |
| | + **RAVR** (Ours) | $34.60_{(4566)}$ | $\mathbf{55.50}_{(2538)}$ | $\mathbf{29.17}_{(6593)}$ | $22.71_{(6834)}$ | $\mathbf{69.69}_{(4615)}$ | $\mathbf{58.43}_{(3261)}$ | 45.05 | **45.00** | **45.02** |

### 3.1 PERFORMANCE OF **RAVR**

We conducted experiments with Qwen3-1.7B (Yang et al., 2025). To validate the generalizability of **RAVR** across different domains, we train the LLM on two different datasets, CrossThink-QA (Akter et al., 2025) and DeepMath-103K (He et al., 2025; Liu et al., 2025). The former is constructed for general purpose reasoning, it includes STEM fields, Economics, Social Sciences, and more. The latter is a math dataset designed with a focus on challenging math problems. We also evaluate the model on datasets from different domains, i.e., general and math benchmarks. For general tasks, we include GPQA-Diamond (Rein et al., 2024) and MMLU-Pro (Wang

et al., 2024); for math, we include AIME24, AIME25, AMC23 and Minerva (Lewkowycz et al., 2022). This also makes it possible to analyze the effectiveness on in-domian and out-of-domain performance. As for baselines, our selection spans two axes: reward type (verifiable reward vs. reference-answer-probability reward) and the presence or absence of implicit curriculum learning to improve stability when learning with challenging samples. The reference-answer-probability reward is technically related to **RAVR**, while implicit curriculum learning is related in terms of our objective of learning from challenging samples. Specifically, we include GRPO (Shao et al., 2024), DAPO (Yu et al., 2025a), VeriFree (Zhou et al., 2025b) and RLPR (Yu et al., 2025b). GRPO and DAPO are widely-acknowledged methods with verifiable reward and DAPO introduces dynamic sampling trick to filter hard examples to achieve curriculum learning. VeriFree and RLPR are up-to-date methods with reference-answer-probability reward and RLPR introduces a reward standard deviation filtering trick to introduce an adaptive curriculum learning. See Appendix A.6 for more implementation details.

The experimental results, summarized in Table 1, demonstrate the superior performance and generalization capabilities of **RAVR**. When trained on the general-purpose CrossThink-QA dataset. Here, **RAVR** achieves the highest overall average with 44.75. It establishes a significant lead on in-domain general tasks with a score of 48.39, outperforming DAPO by a large margin. Crucially, it also demonstrates the best out-of-domain performance on math reasoning tasks (42.92), showcasing its robust ability to transfer learned reasoning skills across domains. Similarly, when trained on the DeepMath dataset, **RAVR** also achieves the highest overall average score of 45.02 and shows strong generalization to out-of-domain general tasks, achieving a leading score of 45.05. **RAVR** consistently delivers state-of-the-art results across both training settings and evaluation benchmarks, validating its effectiveness and robustness. We further examine how the average reasoning length relates to performance. Overall, all RL-based methods substantially improve accuracy over the vanilla model while typically using fewer tokens, and **RAVR** tends to be more *token-efficient*: when trained on CrossThink-QA, **RAVR** saves more tokens than GRPO on 4 out of 6 datasets; when trained on DeepMath, **RAVR** achieves on-par token budget GRPO and uses fewer tokens than DAPO. These observations suggest a reasonable length regime where chains are detailed enough to support correct reasoning but not overly verbose. **RAVR** tends to stay in this regime, likely because answer-conditioned reasoning, guided by the reference answer, prunes redundant or overthinking segments and encourages compact yet effective reasoning during training.

Table 2: Generalizability experiments on more benchmarks.

| Training Set | Model | StrategyQA Avg@4 | TheoremQA Avg@4 | WebInstruct Avg@4 | Olympiad-math Avg@4 | Olympiad-physics Avg@4 | Average |
|---|---|---|---|---|---|---|---|
| CrossThink-QA | + GRPO | 64.25 | 58.04 | 75.52 | 56.23 | **13.14** | 53.44 |
| | + **RAVR** (Ours) | **66.17** | **58.50** | **76.27** | **56.67** | 12.95 | **54.11** |
| DeepMath | + GRPO | 62.75 | 57.33 | 72.67 | 58.17 | 11.97 | 52.58 |
| | + **RAVR** (Ours) | **63.33** | **57.73** | **75.83** | **59.83** | **12.29** | **53.80** |

Table 3: Generalizability experiments on DeepSeek-R1-1.5B.

| Training Set | Model | GPQA-D Avg@4 | MMLU-Pro Avg@4 | AIME 24 Avg@16 | AIME 25 Avg@16 | AMC 23 Avg@16 | Minerva Avg@16 | General Task | Math Reasoning | Average |
|---|---|---|---|---|---|---|---|---|---|---|
| | | In-domain | | Out-of-domain | | | | | | |
| CrossThink-QA | + GRPO | 24.75 | 33.65 | 18.75 | 19.17 | 41.88 | 30.70 | 29.20 | 27.63 | 28.15 |
| | + **RAVR** (Ours) | **25.76** | **33.95** | **20.42** | 18.75 | **42.97** | **31.89** | **29.86** | **28.51** | **28.96** |

Table 4: Generalizability experiments on Qwen3-4B.

| Training Set | Model | GPQA-D Avg@4 | MMLU-Pro Avg@4 | AIME 24 Avg@16 | AIME 25 Avg@16 | AMC 23 Avg@16 | Minerva Avg@16 | General Task | Math Reasoning | Average |
|---|---|---|---|---|---|---|---|---|---|---|
| | | Out-of-domain | | In-domain | | | | | | |
| DeepMath | + GRPO | 50.13 | 67.45 | 49.79 | 38.33 | 86.25 | 68.75 | 58.79 | 60.78 | 60.12 |
| | + DAPO | 50.38 | **68.40** | 47.29 | 37.71 | 85.63 | 67.64 | 59.39 | 59.57 | 59.51 |
| | + **RAVR** (Ours) | **52.65** | 67.88 | **50.21** | **39.79** | **87.19** | **69.12** | **60.27** | **61.58** | **61.14** |

## 3.2 GENERALIZABILITY ANALYSIS OF **RAVR**

To assess the generalizability of **RAVR**, we further evaluate it on a broader set of datasets and models, primarily comparing against GRPO. Overall, **RAVR** yields consistent gains. First, we consider benchmarks that go beyond multiple-choice and standard math problems and instead require stronger compositional reasoning, such as TheoremQA, a theorem-driven QA benchmark that tests the ability to apply formal theorems to solve challenging science problems. The results in Table 2 show that **RAVR** improves over GRPO in most settings. We then test DeepSeek-R1-1.5B trained on CrossThink-QA and compare **RAVR** with GRPO (Table 3). **RAVR** outperforms GRPO on 5 out of 6 datasets and increases the overall average accuracy by about 2.9%. Moreover, we study a larger and stronger model, Qwen3-4B. Since Qwen3-4B already attains nearly 90% accuracy on the CrossThink-QA training distribution, we instead train on the more challenging DeepMath dataset and include DAPO, a strong method for math reasoning, as an additional baseline. As shown in Table 4, **RAVR** consistently surpasses GRPO and further outperforms DAPO on 5 out of 6 datasets, with average improvements of about 1.7% and 2.7%, respectively. These results indicate that the benefits of **RAVR** persist across datasets, domains, and model backbones.

### 3.3 REASONING BEHAVIOR OF **RAVR**

We compare the internal-thinking style of the model trained with **RAVR** and GRPO using frequencies of discourse markers as proxies for cognitive moves (Wang et al., 2025a; Bogdan et al., 2025). The key features of **RAVR** are observed as follows: *(1) Fewer wait.* **RAVR** produces fewer hesitation cues, indicating reduced dithering; this is consistent with stronger problem-solving competence and with answer-conditioned paths that tend to terminate once the reference answer is reached, avoiding unnecessary over-thinking. *(2) More therefore.* This suggests firmer result consolidation: the model more actively reviews preceding steps and commits to a conclusion. *(3) More recall in knowledge QA.* This evidences targeted retrieval aligned with task requirements rather than jumping straight to an answer. *(4) More alternatively and I need in math.* These markers reflect greater

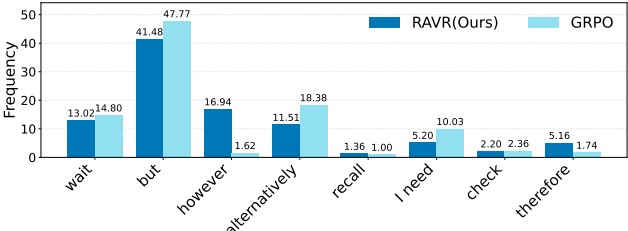

(a) Trained on CrossThink-QA; Tested on GPQA-Diamond.

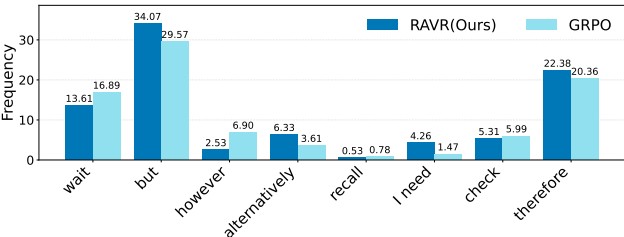

(b) Trained on DeepMath; Tested on AMC23.

Figure 3: Comparison of reasoning behaviors within thinking tags. Words on the x-axis are those frequently used during thinking, and the y-axis represents their average frequency per response. See Appendix A.8 for more results.

divergent exploration and explicit planning before committing to a solution path—desirable for multi-step problem solving like math problem. *(5) Task-adaptive contrast.* In English discourse, *but* and *however* both serve as contrastive markers, yet they differ in usage. *But* is more casual and often signals a local correction or small-scale turn within a sentence, whereas however is more formal and typically marks a global contrast or structured shift across sentences. Multiple-choice task exhibits more *however* (global comparison across options), while math shows more *but* (incremental corrections within derivations), suggesting an adjustment of *contrast granularity* to task demands rather than a fixed stylistic habit. The reasoning behavior shifts indicate a more interpretable, problem-adaptive reasoning process that **RAVR** achieves.

### 3.4 LEARNING DYNAMICS OF **RAVR**

To further investigate how introducing answers enhances sampling efficiency, we compare our model against GRPO with larger rollout group sizes. Figure 4 presents the results on GPQA-Diamond and MMLU-Pro benchmarks, which demonstrate the superior sampling efficiency of our method.

The primary observation is that while the performance of GRPO scales with a larger group size (from 8 to 24), **RAVR** achieves a better or comparable performance with a significantly smaller group size of only 8. This finding provides strong evidence that **RAVR** markedly enhances the sampling efficiency of high-quality reasoning paths. Furthermore, the smoother learning curve for **RAVR** indicates improved learning stability.

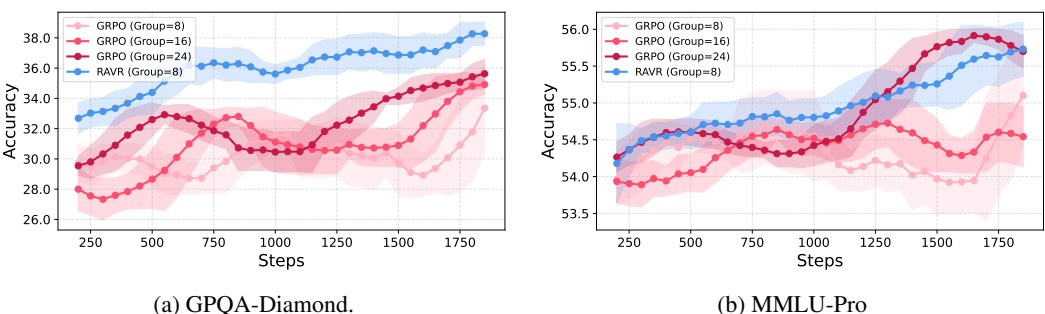

(a) GPQA-Diamond.                    (b) MMLU-Pro

Figure 4: Comparison with GRPO across different rollout group sizes. When using a rollout group size of 8, **RAVR** attains or exceeds the performance of GRPO with a rollout group size of 24. This observation suggests that our approach markedly enhances the sampling efficiency of high-quality reasoning paths, thereby improving learning stability and efficiency.

Moreover, we observe that the KL divergence between $\pi_\theta(z|x, y^*)$ and $\pi_\theta(z|x)$ first fluctuates but then gradually decreases. This indicates that the capability of producing high-utility reasoning paths under the posterior is transferred to the question-only setting as expected, and that the language style of the generated reasoning increasingly aligns with that of reasoning when question-only. Additionally, it's observed that while the prior reasoning utility becomes better, the posterior keep a stable utility gain, which ensures the continual learning.

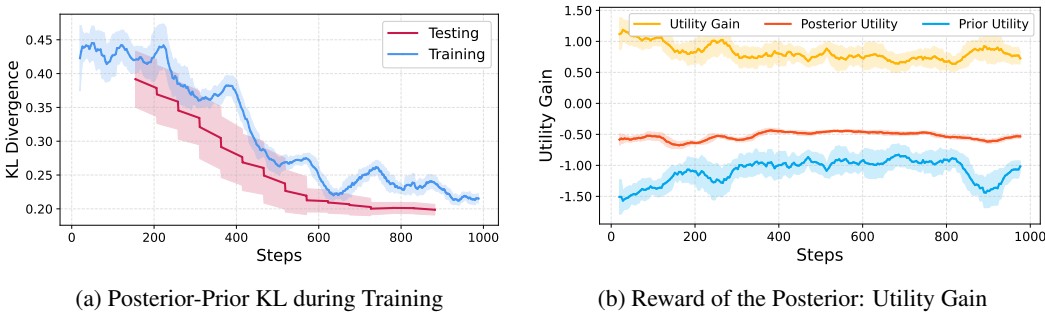

(a) Posterior-Prior KL during Training       (b) Reward of the Posterior: Utility Gain

Figure 5: Learning Dynamics. Training on CrossThink-QA, testing on GPQA and MMLU-pro.

## 3.5 ABLATION STUDY OF **RAVR**

*(i) Main variational reasoning objective.* Maximizing the utility of answer-conditioned reasoning is critical: removing this makes **RAVR** consistently underperform GRPO across both datasets. Without this "utility anchor," the KL term can pull the question-only prior toward a diffused posterior, hindering effective prior learning. Removing the posterior–prior KL term reduces training to an auxiliary reasoning task—producing reasoning given both question and answer. On the math dataset, we even observe slight gains, likely because answers add little beyond flagging errors and avoiding overthinking, making posterior and prior reasoning largely similar. *(ii) Utility baseline and posterior instruction.* Both strategies are important. Without the prior-based utility baseline, posterior reasoning need not surpass prior reasoning, and training drifts toward "least bad" patterns rather than genuinely informative ones. With the baseline, any posterior trace whose induced answer likelihood fails to exceed that of the prior gets zero reward, ensuring updates focus on reasoning that truly improves the prior. Omitting explicit instructions for first-person, "think-aloud" monologues often destabilizes training—likely because the language-style shift disrupts original reasoning patterns. *(iii) KL sample weighting and answer prefix.* These stability-oriented strategies play distinct

roles. KL reweighting has minimal effect on peak accuracy, but removing it causes higher variance. The answer prefix—absent from prior probability-reward methods—helps stabilize early training, enabling faster, more consistent convergence. *(iv) Prior objective.* Retaining the question-only reasoning objective is vital—especially for math—since mismatches between training (posterior) and inference (prior) can degrade a probabilistic model's performance if ignored. Overall, these ablations validate our full design of **RAVR**, achieving both superior stability and higher performance.

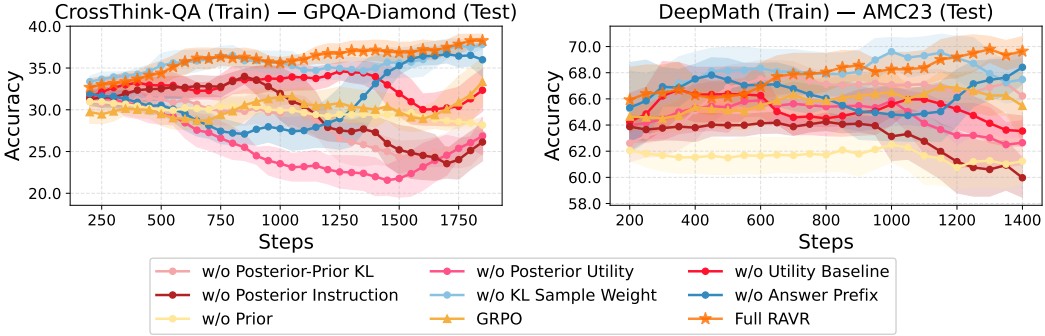

Figure 6: Ablation Study.

## 4 RELATED WORKS

Reinforcement learning has been applied to improve the reasoning abilities of large language models, but it still faces inherent challenges in exploration. Several works have shown that revealing the correct answer to the model can significantly help in generating a useful chain-of-thought, which can then be used for supervised fine-tuning. For instance, Zelikman et al. (2022) introduced STaR, a bootstrapping loop that first let the model attempt each problem on its own, and if it fails, then provides the ground-truth and asks the model to solve again. Recently, Zhou et al. (2025a) expands STaR into the RL setting, which uses STaR to generate positive samples for preference optimization, such as DPO (Rafailov et al., 2023). While these methods demonstrated the power of using answers to obtain high-quality reasoning paths; however, *they are not end-to-end* – it requires interlaced stages of generation and fine-tuning. Moreover, they judge generated chain-of-thoughts purely by whether they end in the correct answer, which may not capture other important qualities of an effective reasoning chain. Another related work comes from the teacher-student distillation paradigm. Cetin et al. (2025) trains a teacher model, *which requires a complete step-by-step solution of a problem as input and instantly outputs an explanation, without engaging in any reasoning.* This yields very detailed and instructive explanations that can be used to distill stronger students. However, this work ignores the capacity of LLMs for reasoning in explanation derivation and it adopts an elaborate two-model, two-phase setup, making it a complex approach.

In contrast to these approaches, **RAVR** incorporates the reference answer into an end-to-end training framework to directly improve the LLM's own problem-solving and reasoning abilities. Notably, **RAVR** does not require the reference answer to be a complete solution—even a single word or phrase can suffice—since it fully exploits the LLM's inherent reasoning capability.

## 5 CONCLUSION

This work presents **RAVR**, an end-to-end framework that makes the LLM think *why is this the answer* to help its learning of *what is the answer* with a novel variational reinforcement learning objective. This is the first work that leverages the reasoning ability of LLMs to utilize the reference answer to derive high-quality reasoning paths. We further innovatively propose simple but effective strategies, i.e., the reasoning reward baseline, the reward-based sample weight in KL calculation and the answer prefix trick, to enhance the training stability and performance. In the future, we will explore the open-ended tasks, where the reference answers offer richer information and it is typically more difficult for LLMs to obtain high-quality reasoning solely through their inherent capabilities.

## ETHICS STATEMENT

This work does not involve human subjects, sensitive personal data, or biased decision-making. The main potential negative impact is the carbon footprint associated with the computational resources used in model training and inference. We have taken measures to reduce environmental impact, e.g., using efficient training and inference techniques.

## REPRODUCIBILITY STATEMENT

We have undertaken several measures to ensure the reproducibility of our work. The implementation code of **RAVR** is provided in the supplementary materials with the environment requirements, and detailed descriptions of the experimental settings, including hyperparameter configurations, are given in Appendix A.6. All training and test datasets used in our experiments are publicly available, with their sources clearly cited in the main text. Furthermore, all mathematical results in this paper are accompanied by precise definitions and complete derivations, which can be found in sections such as Section 2.2, as well as in Appendix A.4 and Appendix A.5.

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

# A APPENDIX

## A.1 THE USE OF LARGE LANGUAGE MODELS (LLMS)

In the course of this work, Large Language Models (LLMs) were employed as auxiliary tools in the following ways:

1. **Figure generation** — The model *Gemini-2.5-Pro* (with *Nano Banana*) was used to assist in the creation of Figure 1.

2. **Text editing** — *GPT-5* was utilized for grammar checking, paraphrasing, and shortening of sections of the manuscript.

3. **Code assistance** — *VSCode Copilot* powered by *GPT-5* was used to aid in code implementation, particularly for visualization-related scripts.

## A.2 MOTIVATION EXPERIMENTS

We used Qwen3-1.7B with thinking mode enabled and ran inference using the officially recommended settings: Temperature=0.6, TopP=0.95, TopK=20; the maximum completion tokens were set to 8192.

One experimental case is as follows. The question asked: *Which renewable energy source has the highest capacity factor, making it the most reliable for consistent power generation?* The correct answer is *geothermal energy*. When only the question was provided, the model predicted *hydroelectric energy*, overestimating hydro's capacity factor by assuming that water is always flowing and overlooking its dependence on water availability—some plants are used for peak power, so actual average can be lower. It also failed to analyze geothermal in depth after deeming *hydropower* the correct answer and arrived at an incorrect, intuition-driven estimate of geothermal's capacity factor.

When the reference answer was provided, the model noted that *hydroelectric* output can vary with location and season, and it carefully reasoned that geothermal plants can operate continuously as long as there is a viable geothermal resource; once the plant is running, it can produce steady power. It also gave more accurate estimates of the capacity factors of hydroelectric and geothermal sources. As a result, GPT-5 evaluated the derived reasoning as: *It correctly defines capacity factor, compares sources, and concludes geothermal has the highest capacity factor and reliability; minor oversimplifications (e.g., biomass) do not affect the correct conclusion. The reasoning is self-contained, does not reference the provided answer, and shows no signs of relying on it.*

Overall, *when given the reference answer, the model did not collapse into answer-conforming shortcuts. On the contrary, the presence of the answer promoted a more detailed discussion of both the initially uncertain (incorrect) option and the correct one.*

## Case Study of Motivation Experiment

**A. Question; Reference Answer of This Question is D.**
Which of the following renewable energy sources has the highest capacity factor, making it the most reliable for consistent power generation?
A: Solar photovoltaic (PV) energy
B: Wind energy
C: Hydroelectric energy
D: Geothermal energy
E: Biomass energy
F: Tidal energy
G: Wave energy
H: Solar thermal energy
I: Biofuels
J: Hydrogen fuel cells
Please think step by step and output the final answer within \boxed{}.

**B. Reasoning and Final Response When Question-only; Using Prompt in Section 2.3.**
Okay, let's try to figure out which renewable energy source has the highest capacity factor. First, I need to remember what capacity factor is. From what I recall, capacity factor is the ratio of actual output over the maximum possible output. So, it's like how much of the potential energy a power plant can produce in a given time. A higher capacity factor means the energy source is more reliable because it produces more consistently.

Now, looking at the options: A to J. Let me go through each one.

Starting with Solar photovoltaic (PV) energy. I think solar PV depends on sunlight, which is variable. Even on cloudy days, it might not produce as much. So maybe its capacity factor is lower compared to others. But I'm not sure. I remember that solar panels have a capacity factor around 15-20% for PV, depending on location and weather. Maybe that's not the highest.

Wind energy. Wind power is also variable because wind doesn't blow all the time. The capacity factor for wind is around 30-40% in some places, but it can vary. For example, in areas with consistent wind, like coastal regions, it might be higher. But I think wind's capacity factor is lower than hydroelectric. Wait, but maybe in some places it's higher?

Hydroelectric energy. Hydroelectric plants have a high capacity factor because they use the water flow from rivers, which is more consistent. They can generate electricity 24/7 as long as there's water flow. I think hydroelectric has a capacity factor around 80-90% in many places. That seems really high. But wait, does that apply to all hydroelectric plants? Maybe some are more efficient, but generally, hydro is pretty reliable.

Geothermal energy. Geothermal plants can be pretty reliable because they use the Earth's internal heat. But their capacity factor depends on the location. For example, in places with consistent geothermal activity, like Iceland or the US, they might have a high capacity factor. But I think it's around 50-60% maybe? Not sure. Maybe lower than hydro.

Biomass energy. Biomass is from organic materials. It's variable because it depends on the availability of biomass. For example, if there's a lot of crops or waste, it can be more reliable. But the capacity factor for biomass is usually lower than hydro. Maybe around 20-30%?

Tidal energy. Tidal energy depends on the tides, which are predictable. So maybe it has a higher capacity factor? But I think it's lower than hydro because tides are less consistent than river flows. But I'm not sure. Tidal energy's capacity factor might be around 50-60%, maybe?

Wave energy. Similar to tidal, but wave energy is more variable. The capacity factor for wave energy is probably lower than tidal. Maybe around 20-30%?

Solar thermal energy. Solar thermal uses sunlight to generate heat, which then produces electricity. But it's similar to solar PV in that it depends on sunlight. So maybe similar capacity factor as PV, around 15-20%.

Biofuels. Biofuels are made from organic matter, but they can be less reliable because they depend on agricultural inputs. Their capacity factor might be lower than hydro. Maybe around 20-30%?

Hydrogen fuel cells. Hydrogen is produced via electrolysis, which is usually from renewable sources. But hydrogen fuel cells themselves are not a power source. Wait, the question is about renewable energy sources. So hydrogen fuel cells are not a source of energy but a method of storing energy. So maybe not part of the options here. Wait, the options include "Hydrogen fuel cells" as an option. But the question is about renewable energy sources, so maybe hydrogen is considered a renewable source, but the fuel cell is the storage. So maybe the answer is not here. But the question says "renewable energy sources", so hydrogen fuel cells are not a source, but a method. So maybe not part of the options. Wait, but the options are all renewable energy sources. Wait, the options are: A to J. So maybe hydrogen fuel cells are considered a renewable energy source? Or is it not? Hmm. Maybe not. So perhaps the answer is hydroelectric.

So, going back. Hydroelectric has the highest capacity factor, like 80-90% in some places. Wind is around 30-40%, solar PV 15-20%, geothermal 50-60%, tidal 50-60%, wave 20-30%, biomass 20-30%, biofuels 20-30%, hydrogen fuel cells... not sure. So the highest is hydroelectric. So the answer would be C: Hydroelectric energy.

But wait, I need to check if there's any other sources with higher capacity factors. For example, in some regions, wind might have higher capacity factors. But globally, hydroelectric is the highest. Also, the question says "the most reliable for consistent power generation". Hydroelectric is the most reliable because it's based on water flow, which is more consistent than wind or solar. So yes, I think the answer is C.
</think>

To determine which renewable energy source has the highest capacity factor, we must first understand what capacity factor means. It is defined as the ratio of actual energy output over the maximum possible output of a power plant, typically expressed as a percentage. A higher capacity factor indicates that a power source is more reliable and consistent in producing energy over time.

—

Key Renewable Energy Sources and Their Capacity Factors
Let's evaluate each option:
- A: Solar photovoltaic (PV) energy
- Capacity factor: Typically 15–20%
- Variability: Dependent on sunlight, which is limited by weather and time of day.
- B: Wind energy
- Capacity factor: 30–40% in some locations, but can drop significantly in low-wind areas.
- Variability: Wind is unpredictable, so capacity factor is lower than hydroelectric.
- C: Hydroelectric energy
- Capacity factor: 80–90% in many regions, especially where there is consistent water flow (e.g., rivers, dams).

- Reliability: Water flow is predictable and consistent, making it one of the most reliable sources.
- D: Geothermal energy
- Capacity factor: 50–60% in regions with stable geothermal activity.
- Variability: Relatively stable but not as consistent as hydroelectric.
- E: Biomass energy
- Capacity factor: 20–30%.
- Variability: Relies on biomass availability, which can be limited.
- F: Tidal energy
- Capacity factor: 50–60%.
- Variability: Tides are predictable but less consistent than river flows.
- G: Wave energy
- Capacity factor: 20–30%.
- Variability: Less predictable and more variable than tidal or hydroelectric.
- H: Solar thermal energy
- Capacity factor: 15–20%.
- Similar to solar PV in terms of variability.
- I: Biofuels
- Capacity factor: 20–30%.
- Variability: Dependent on agricultural inputs.
- J: Hydrogen fuel cells
- Not a source of energy but a form of energy storage.
- Not considered a renewable energy source in this context.
—
Conclusion
Among all the listed options, hydroelectric energy stands out as the most reliable and consistent source of power due to its predictable and stable energy output. It has the highest capacity factor, typically 80–90%, and is not subject to the same variability as solar or wind energy.
—
Final Answer
\boxed{C}

**C. Deriving Valid Reasoning with Reference Answer; Using Prompt in Section 2.3.**
Okay, let's tackle this question. The user is asking which renewable energy source has the highest capacity factor, making it the most reliable for consistent power generation. The options are from A to J, *and the reference answer is D, which is geothermal energy. But I need to figure this out step by step without referencing the answer.*
First, I need to recall what capacity factor means. Capacity factor is the ratio of actual output to the maximum possible output over a given period. A higher capacity factor means the energy source can produce more consistent power. So, the question is about which source can produce the most energy on average, even if it's not always producing at maximum capacity.
Let me go through each option one by one. Starting with solar PV. I know that solar energy depends on sunlight, which is intermittent. It's not always sunny, so the capacity factor is lower. For example, solar PV might have a capacity factor around 20-30% depending on location and weather.
Next, wind energy. Wind is also intermittent. The capacity factor for wind varies, but it's usually around 30-40% in some regions. However, wind turbines can be more variable because wind speeds fluctuate. But I remember that in some areas, wind farms have higher capacity factors. But I need to check if that's the case.
Hydroelectric energy. This is more reliable. Hydropower plants can generate electricity when there's water flow, which is typically consistent. However, *the capacity factor for hydro is around 40-60% depending on the location and the season.* But wait, some hydro plants have higher capacity factors because they can store water and release it when needed. But I need to confirm if that's accurate.
Geothermal energy. This is a bit more unique. *Geothermal plants can operate continuously as long as there's geothermal activity. The capacity factor for geothermal is usually high, maybe around 60-70%? Because once the plant is running, it can produce consistently. But I'm not sure about the exact numbers.*
Biomass energy. This is a bit tricky. Biomass can be intermittent because it depends on agricultural waste or organic materials. The capacity factor might be lower compared to others. Also, biomass can be more variable depending on the feedstock and availability.
Tidal energy. Tidal energy is predictable because tides are regular. But the capacity factor might be lower than hydro because the energy is tied to the tides, which are not as consistent as the river flow. However, some tidal plants have high capacity factors.
Wave energy. Similar to tidal, but wave energy is more variable. The capacity factor might be lower than tidal, but I'm not sure.
Solar thermal energy. This is a type of solar power that uses concentrated sunlight. It's similar to solar PV but can have higher efficiency. However, the capacity factor might be similar to solar PV because it's still dependent on sunlight availability.
Biofuels. These are derived from organic matter. They can be intermittent if the feedstock is not available, but in some cases, they can be more consistent. However, the capacity factor might be lower than others.
Hydrogen fuel cells. These are a type of energy storage, not a power generation source. So they might not have a high capacity factor because they are used for storage, not generation. Wait, but the question is about renewable energy sources for power generation. Hydrogen fuel cells are more about storage, so maybe they are not considered here. So maybe they are not in the options. Wait, the options include J: Hydrogen fuel cells. But the question is about renewable energy sources for power generation. So hydrogen fuel cells might be considered as a source of energy, but their capacity factor is lower because they require storage. So maybe they are not the best answer.
So, going back. The options are A to J. *The reference answer is D: geothermal. But I need to figure out why.*
Hydroelectric is a good candidate because it's consistent. But geothermal is even better because it's a continuous source. However, I need to check if geothermal is the highest capacity factor.
Another point: geothermal plants can operate 24/7, but they require specific geological conditions. So their capacity factor is high. For example, in some regions, geothermal plants have a capacity factor over 80-90%? Wait, that seems high. But I think that's correct. Because once the plant is running, it can produce consistently. So geothermal has a very high capacity factor.
So the reasoning would be: solar PV is intermittent, wind has variable output, hydro is consistent but maybe not as high as geothermal. Geothermal is the most reliable because it's continuous. So the answer is D.
But I need to make sure there's no other option with a higher capacity factor. Let me check again.
Tidal energy is predictable but maybe not as high as geothermal. Wave energy is similar. Solar thermal is similar to solar PV. Biofuels and hydrogen are less reliable. So geothermal is the best.
Therefore, the answer is D.
</think>

I begin by defining the term "capacity factor" as the ratio of actual energy output to the maximum possible output over a given period. This metric is crucial for assessing the reliability of a renewable energy source. My goal is to determine which option provides the highest capacity factor, ensuring consistent power generation.

Starting with solar photovoltaic (PV) energy, I recognize that its output depends on sunlight availability, which is intermittent. This leads to a lower capacity factor, typically around 20–30%, depending on geographic and meteorological conditions. I note that solar PV is not a constant source, making it less reliable for consistent power generation.

Next, wind energy. While wind is a viable option, its capacity factor is influenced by variable wind speeds. In regions with consistent wind patterns, the capacity factor can reach 30–40%, but this is still less than hydroelectric sources. I consider that wind's variability limits its reliability compared to more stable energy sources.

Hydroelectric energy stands out due to its dependence on water flow, which is influenced by seasonal and climatic factors. However, hydroelectric plants often have higher capacity factors, typically ranging from 40–60%, especially in regions with consistent river flow. I acknowledge that while hydro is reliable, its capacity factor is lower than geothermal.

Geothermal energy is unique in that it operates continuously as long as there is geothermal activity. This ensures a high capacity factor, often exceeding 60–70%, depending on the site. The consistency of geothermal energy is unmatched by other sources, making it a strong candidate for reliability.

Biomass energy and hydrogen fuel cells are less reliable due to their dependence on feedstock availability and storage. Biomass is intermittent, and hydrogen fuel cells require storage infrastructure, which complicates their capacity factor. These options are not as consistent as geothermal.

Tidal and wave energy are predictable but have lower capacity factors compared to hydroelectric. Tidal energy, for example, is limited by the periodicity of tides, which results in a lower capacity factor than hydroelectric.

I reassess the options, considering the variability of each energy source. While hydroelectric and geothermal both have high capacity factors, geothermal's continuous operation and stable output make it the most reliable. I verify that geothermal's capacity factor is indeed the highest among the listed options, ensuring that my reasoning is comprehensive and accurate.

Thus, the most reliable renewable energy source for consistent power generation is geothermal energy.

**D. GPT-5 Judgement.**
**Prompt**
Please evaluate the following reasoning process on two aspects:
Question: {{question}}
Reasoning Process: {{derived reasoning process}}
Reference Answer: {{reference answer}}
Please evaluate separately:
1. Correctness Assessment: Does this reasoning process correctly lead to the answer?
- Judge whether the reasoning logic is correct
- Judge whether each step is reasonable and coherent
- Judge whether the final conclusion is correct
2. Independence Assessment: Does this reasoning process not rely on the reference answer?
- Judge whether there are obvious traces of using the reference answer
- Judge whether there are statements like "I know the answer is X"
Please answer in the following format strictly:
Correctness Assessment: Yes/No
Correctness Explanation: one or two concise sentences
Independence Assessment: Yes/No
Independence Explanation: one or two concise sentences

**Judgement**
Correctness Assessment: Correctness Assessment: Yes
Correctness Explanation: It correctly defines capacity factor, compares sources, and concludes geothermal has the highest capacity factor and reliability; minor oversimplifications (e.g., biomass) do not affect the correct conclusion.

Independence Assessment: Yes
Independence Explanation: The reasoning is self-contained, does not reference the provided answer, and shows no signs of relying on it.

## A.3 RATIONALITY ANALYSIS OF ANSWER-CONDITIONED REASONING

We further analyze the quality of answer-conditioned reasoning. In this analysis, we randomly sample 500 examples from the CrossThink-QA dataset. For each example, we enable the Qwen3-1.7's thinking mode and generate 8 question-only prior reasoning paths. We then retain only those questions for which none of the 8 prior reasoning paths produces the correct answer, yielding a set of 85 challenging questions in total. For each of these 85 questions, the model also produces up to 8 answer-conditioned posterior reasoning paths.

The answer-conditioned reasoning paths can be categorized into four types based on whether the reasoning is logically valid (i.e., free of hallucinations or reward hacking) and whether it is independent of the reference answer (i.e., does not indicate prior access to the reference answer): *valid and independent*, *valid and dependent*, *invalid and independent*, and *invalid and dependent*. We use GPT-5.1 to annotate each reasoning path with the same prompt in part D in the above case study. In our data with the raw model weights, valid and independent reasoning paths appear on 62.4% of

| Reasoning type | $n$ | mean | std | min | 25% | 50% | 75% | max |
|---|---|---|---|---|---|---|---|---|
| Valid & Independent | 53 | 4.97 | 4.25 | -7.18 | 3.63 | 5.45 | 7.41 | 12.38 |
| Valid & Dependent | 31 | 4.11 | 4.36 | -6.23 | 0.54 | 5.06 | 7.29 | 11.03 |
| Invalid & Independent | 62 | 2.09 | 3.93 | -7.71 | -0.72 | 1.20 | 5.16 | 10.80 |
| Invalid & Dependent | 59 | 3.27 | 4.15 | -6.38 | -0.04 | 3.97 | 5.95 | 13.65 |

Table 5: Per-example improvement $\Delta \log p$ for answer-conditioned reasoning types. Here $n$ is the number of examples that contain at least one reasoning path of the given type; statistics are computed over the per-example means. Valid and independent reasoning achieves the largest mean and median gains, with a notably high 25th percentile (3.63), indicating strong and stable improvements over the question-only baseline and other reasoning types at the distribution level.

| Comparison | $n$ | $t$ | $p$ |
|---|---|---|---|
| Valid & Independent vs. Valid & Dependent | 28 | -1.54 | 0.134 |
| Valid & Independent vs. Invalid & Independent | 34 | 2.93 | 0.006 |
| Valid & Independent vs. Invalid & Dependent | 32 | 2.15 | 0.039 |

Table 6: Paired $t$-tests on per-example $\Delta \log p$, comparing valid and independent reasoning against other answer-conditioned reasoning types. Each test is computed over the subset of examples where both reasoning types are present.

examples and account for 32.1% of answer-conditioned trajectories per example on average. For each reasoning path we measure the log-probability of the ground-truth answer, and for each example we compute the mean over reasoning paths of the same type. Empirically, we find that *valid reasoning paths lead to substantially higher ground-truth answer probabilities than invalid ones*, even though most of the invalid reasoning paths in our analysis also end with the correct answer. In other words, reasoning paths that contain hallucinations or reward hacking are naturally assigned lower reward under group-relative reward normalization and are suppressed during training.

Specifically, the analysis is as follows. For each example and each reasoning type that is present on that example, we compute the improvement

$$\Delta \log p = \log p(\text{reference answer} \mid \text{reasoning type}) - \mathbb{E}[\log p(\text{reference answer} \mid \text{question-only})],$$

where $\mathbb{E}[\log p(y^\star \mid \text{question-only})]$ is the mean log-probability of the ground-truth answer $y^\star$ over the eight question-only reasoning paths. Table 5 summarizes the distribution of $\Delta \log p$ across examples for each reasoning type. All four types yield positive mean improvements over the question-only baseline, but valid and independent reasoning achieves the largest mean and median gains, with a particularly high 25th percentile (3.63), indicating both strong and stable improvements.

To compare reasoning types on the same examples, we run paired $t$-tests between valid and independent reasoning and each of the other three types, restricted to examples where both types are present (Table 6). Valid and independent reasoning significantly outperforms both invalid types in terms of $\Delta \log p$ ($p < 0.05$), while the difference between valid and independent reasoning and valid but dependent reasoning is not statistically significant. For those reasoning paths that are logically valid but dependent, our qualitative inspection reveals a characteristic failure mode: the final explanation explicitly refers to the provided answer, using phrases such as "according to the reference answer" or "the reference answer is ...". To discourage such behavior, we introduce an additional reward term that penalizes explicit reference-answer leakage. Concretely, if a trajectory contains the substring "reference answer", we assign an extra penalty of $-0.5$ to its reward. This simple heuristic effectively separates valid and independent reasoning from valid but dependent reasoning in practice.

The evolution of reference-answer leakage during training is shown in Figure 7. As training progresses, the proportion of reasoning paths that explicitly mention the reference answer steadily de-

creases. At inference time, where the model does not receive the reference answer, we do not observe such leakage or related hallucinations in the generated reasoning.

## A.4 DECOMPOSITION AND REWRITING OF $\Pr(Z \mid x, y^*)$

$x$ denotes the input problem and $y^*$ the reference answer. $\mathcal{Z}$ is the (countable) set of all reasoning paths; $Z \subseteq \mathcal{Z}$ is a subset of reasoning paths. $\pi_\theta(z \mid x)$ is the model's distribution over reasoning paths given $x$; $\pi_\theta(y^* \mid x, z)$ is the conditional distribution over reference answer given $(x, z)$. Define

$$s(z) := \pi_\theta(y^* \mid x, z), \qquad \mu := \pi_\theta(y^* \mid x).$$

The posterior over paths given $(x, y^*)$ induces a probability on the event $Z$, i.e., sampling a reasoning path that belongs to $Z$:

$$\Pr(Z \mid x, y^*) = \sum_{z \in Z} \pi_\theta(z \mid x, y^*). \tag{16}$$

Apply Bayes' rule, for each $z \in \mathcal{Z}$, Bayes' rule gives

$$\pi_\theta(z \mid x, y^*) = \frac{\pi_\theta(y^* \mid x, z)\,\pi_\theta(z \mid x)}{\pi_\theta(y^* \mid x)} = \frac{s(z)\,\pi_\theta(z \mid x)}{\mu} \quad \text{(assuming } \mu > 0\text{).} \tag{17}$$

Substituting equation 17 into equation 16 yields

$$\Pr(Z \mid x, y^*) = \sum_{z \in Z} \frac{s(z)\,\pi_\theta(z \mid x)}{\mu} = \frac{1}{\mu} \sum_{z \in Z} s(z)\,\pi_\theta(z \mid x). \tag{18}$$

Meanwhile, for a discrete space, the conditional expectation of $s(z)$ given $z \in Z$ (under $\pi_\theta(\cdot \mid x)$) is

$$\mathbb{E}[s(z) \mid z \in Z,\, x] = \sum_{z \in Z} s(z)\,\frac{\pi_\theta(z \mid x)}{\Pr(Z \mid x)} \quad \text{(defined when } \Pr(Z \mid x) > 0\text{),} \tag{19}$$

where by definition,

$$\Pr(Z \mid x) = \sum_{z \in Z} \pi_\theta(z \mid x). \tag{20}$$

Multiplying both sides of equation 19 by $\Pr(Z \mid x)$ gives

$$\sum_{z \in Z} s(z)\,\pi_\theta(z \mid x) = \Pr(Z \mid x) \cdot \mathbb{E}[s(z) \mid z \in Z,\, x]. \tag{21}$$

Substituting equation 21 into equation 18 yields the desired factorization:

$$\Pr(Z \mid x, y^*) = \Pr(Z \mid x) \cdot \frac{\mathbb{E}[s(z) \mid z \in Z,\, x]}{\mu}. \tag{22}$$

## A.5 ELBO DERIVATION WITH THE AMORTIZED ANSWER-CONDITIONED POSTERIOR

$\pi_\theta(z \mid x)$ is the prior over reasoning paths given $x$, and $\pi_\theta(y^* \mid x, z)$ is the likelihood of the reference answer $y^*$ under path $z$. The original optimization objective is maximizing the utility under the reasoning distribution as

$$\log \mathcal{J}(\theta) = \log \mathbb{E}_{z \sim \pi_\theta(z \mid x)} \big[ \pi_\theta(y^* \mid x, z) \big]. \tag{23}$$

Now we introduce an amortized posterior $\pi_\theta(z \mid x, y^*)$ to aid learning of $\pi_\theta(z \mid x)$. Starting from equation 23, multiply and divide inside the expectation by $\pi_\theta(z \mid x, y^*)$, and change the sampling distribution, we have:

$$\log \mathcal{J}(\theta) = \log \mathbb{E}_{z \sim \pi_\theta(z \mid x)} \left[ \pi_\theta(y^* \mid x, z) \cdot \frac{\pi_\theta(z \mid x, y^*)}{\pi_\theta(z \mid x, y^*)} \right]$$

$$= \log \mathbb{E}_{z \sim \pi_\theta(z \mid x, y^*)} \left[ \frac{\pi_\theta(y^* \mid x, z)\,\pi_\theta(z \mid x)}{\pi_\theta(z \mid x, y^*)} \right]. \tag{24}$$

Applying Jensen's inequality to the concavity of natural logarithm function $\log(\cdot)$,

$$\log \mathcal{J}(\theta) \geq \mathbb{E}_{z \sim \pi_\theta(z|x,y^*)}[\log \pi_\theta(y^* \mid x, z) + \log \pi_\theta(z \mid x) - \log \pi_\theta(z \mid x, y^*)] \tag{25}$$

$$= \mathbb{E}_{z \sim \pi_\theta(z|x,y^*)}[\log \pi_\theta(y^* \mid x, z)] + \underbrace{\mathbb{E}_{z \sim \pi_\theta(z|x,y^*)}\left[\frac{\log \pi_\theta(z \mid x)}{\log \pi_\theta(z \mid x, y^*)}\right]}_{\text{forms a } -\text{KL term}} \tag{26}$$

$$= \mathbb{E}_{z \sim \pi_\theta(z|x,y^*)}\left[\log \pi_\theta(y^* \mid x, z)\right] - \mathbb{D}_{\mathrm{KL}}[\pi_\theta(z \mid x, y^*) \,\|\, \pi_\theta(z \mid x)]. \tag{27}$$

Equation equation 27 is the Evidence Lower Bound (ELBO) on $\log \mathcal{J}(\theta)$ with the amortized answer-conditioned posterior $\pi_\theta(z|x, y^*)$. The KL estimator introduced by Schulman (2020) is as follows:

$$\mathbb{D}_{\mathrm{KL}}\left[\pi_\theta(z|x,y^*) \| \pi_\theta(z|x)\right] \approx \frac{\pi_\theta(z_t \mid x, z_{<t})}{\pi_\theta(o_{i,t} \mid x, y^*, \tilde{z}, z_{<t})} - \log \frac{\pi_\theta(z_t \mid x, z_{<t})}{\pi_\theta(z_t \mid x, y^*, \tilde{z}, z_{<t})} - 1. \tag{28}$$

## A.6 IMPLEMENTATION DETAILS

Each experiment is trained on 32 NVIDIA H100 GPUS. For **RAVR**, GRPO and DAPO, the learning rate for the policy model is 1e-6. In each rollout step, we sample eight responses per prompt for a batch of 32 prompts using a temperature of 1, and subsequently perform 2 policy updates on the collected responses with a batch size of 128. We adopt the clip-higher strategy and set the clip threshold as 0.8 and 1.27. For VeriFree and RLPR, we use the default setting in their official code. During evaluation, we follow the official recommended setting of Qwen3-1.7B, i.e., set Temperature=0.6, TopP=0.95 and TopK=20. In the evaluation, we use GPT-4.1-mini for correctness judgement. To reduce the evaluation variance, we report the final Avg4 for general reasoning multiple-choice tasks and Avg4 for math reasoning tasks. The max generation length for training and evaluation is 8192. For DeepMath-103K, we randomly selected 5,000 samples for the training set. For MMLU-pro, we used the subset of the original data containing 1,000 randomly sampled samples (Yu et al., 2025b).

## A.7 SUPPLEMENTARY LEARNING DYNAMICS

We continuously monitor whether the model's responses contain explicit mentions of the phrase "reference answer" We find that, as training progresses, reasoning paths generated under reference-answer conditioning reveal the reference answer less frequently. Meanwhile, in the question-only setting, the model consistently maintains normal output behavior and never produces phrases such as "reference answer."

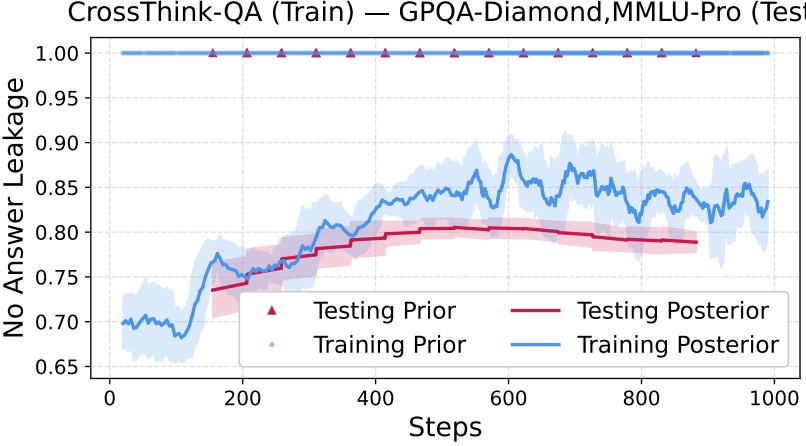

Figure 7: No Reference Answer Leakage

## A.8 MORE REASONING BEHAVIOR RESULTS

The reasoning behavior statistics on other datasets are as follws.

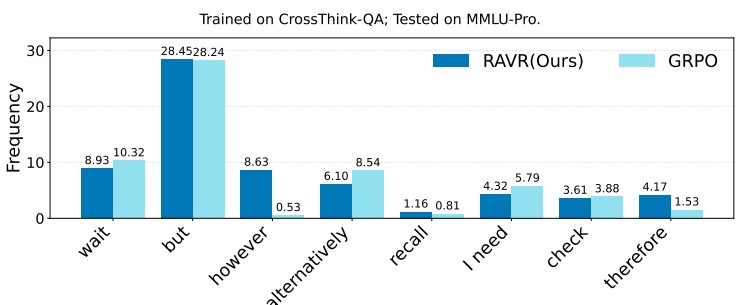

Figure 8: Reasoning Behavior on MMLU

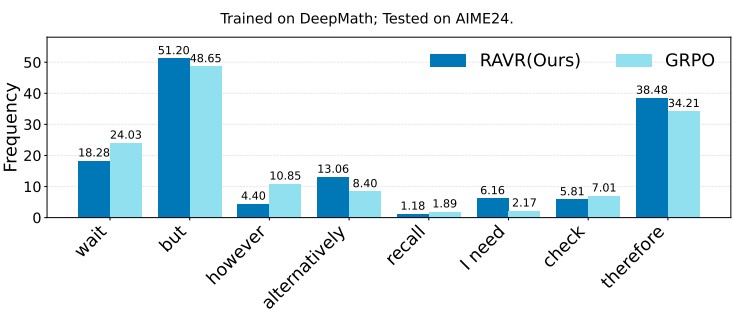

Figure 9: Reasoning Behavior on AIME24

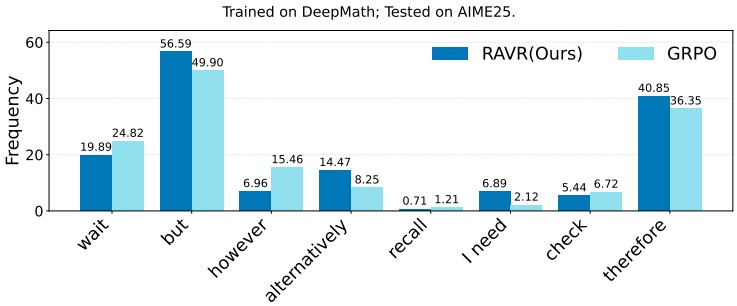

Figure 10: Reasoning Behavior on AIME25

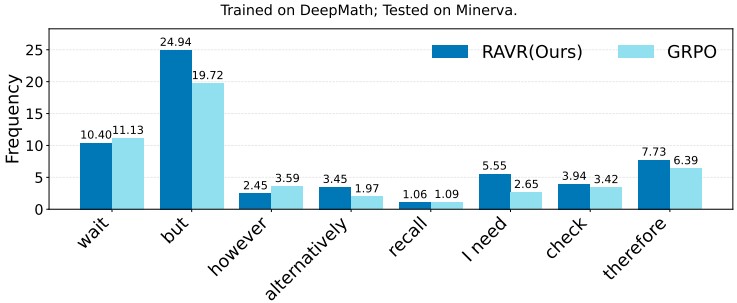

Figure 11: Reasoning Behavior on Minerva

