# OpenReview forum: "RAVR: Reference-Answer-guided Variational Reasoning for Large Language Models"
_ICLR.cc/2026/Conference — ICLR 2026 Conference Withdrawn Submission_

### Official Review · Reviewer_1Zeu · 2025-10-31

**Soundness:** 3
**Presentation:** 2
**Contribution:** 2
**Rating:** 4
**Confidence:** 2

**Summary:**

This paper proposes RAVR (Reference-Answer-Guided Variational Reasoning), an end-to-end framework that leverages reference answers to enhance large language models’ (LLMs) reasoning capabilities in reinforcement learning (RL). It formally proves that conditioning on reference answers increases the expected utility of sampled reasoning paths, transforming intractable tasks into learnable ones by drawing on the cognitive science insight that explaining answers is easier than generating them.

**Strengths:**

solid theoretical foundation, as it provides rigorous mathematical proofs to validate that reference-answer conditioning amplifies high-utility reasoning paths

Its innovative integration of variational inference, using answer-conditioned reasoning as a surrogate for question-only reasoning, enables end-to-end training, avoiding the multi-stage complexity of existing methods like STaR.

**Weaknesses:**

- The reliance on specific LLMs (Qwen3-1.7B) and benchmarks raises concerns about generalizability

- the paper focuses on tasks with clear reference answers (e.g., multiple-choice, math problems), leaving untested whether RAVR works for scenarios where answers are ambiguous or multi-faceted.

- I'm not quite sure how the method performs RL training, e.g., GRPO. Is it that the input is x and y, and the output is z?
What are the reward signals in GRPO? As far as I know, the thought path (z) is not a verifiable signal.

- Additionally, during test-time, since there is no ground truth, how does it perform inference?

**Questions:**

see Weaknesses

---

> ### Author Response · Authors · 2025-11-27
> **Rebuttal by Author (1/3)**
>
> We appreciate your positive assessment of our work’s solid theoretical foundation and its innovative integration of variational inference with answer-conditioned reasoning. Below, we first summarize your main concerns and then respond to each, with a brief TL;DR followed by a more detailed discussion.
>
> ---
>
> ## 1. experiments on more benchmarks and LLMs.
>
> **TL;DR.**
> We added a new section **3.2 Generalizability Analysis of RAVR** with (i) *five more diverse benchmarks* beyond standard multiple-choice/math and (ii) *additional model families and sizes*: DeepSeek-R1-Distill-1.5B and Qwen3-4B. Across all these settings, RAVR consistently improves over strong RL baselines, suggesting that the method generalizes across datasets, domains, and architectures.
>
>
> **Generalizability experiments on more benchmarks.**
>
> | Training Set  | Model           | StrategyQA (Avg@4) | TheoremQA (Avg@4) | WebInstruct (Avg@4) | Olympiad-Math (Avg@4) | Olympiad-Physics (Avg@4) | Average |
> |--------------|-----------------|--------------------|--------------------|----------------------|------------------------|---------------------------|---------|
> | CrossThink-QA | + GRPO          | 64.25              | 58.04              | 75.52                | 56.23                  | **13.14**                     | 53.44   |
> |  | + RAVR (Ours)  | **66.17**              | **58.50**              | **76.27**                | **56.67**                  | 12.95                     | **54.11**   |
> | DeepMath      | + GRPO          | 62.75              | 57.33              | 72.67                | 58.17                  | 11.97                     | 52.58   |
> |      | + RAVR (Ours)   | **63.33**              | **57.73**              | **75.83**                | **59.83**                  | **12.29**                     | **53.80**   |
>
> **Generalizability experiments on DeepSeek-R1-1.5B.**
>
>
> | Training Set   | Model           | GPQA-D (Avg@4) | MMLU-Pro (Avg@4) | AIME 24 (Avg@16) | AIME 25 (Avg@16) | AMC 23 (Avg@16) | Minerva (Avg@16) | General Task | Math Reasoning | Average |
> |---------------|-----------------|----------------|------------------|------------------|------------------|-----------------|------------------|--------------|----------------|---------|
> | CrossThink-QA | + GRPO          | 24.75          | 33.65            | 18.75            | **19.17**            | 41.88           | 30.70            | 29.20        | 27.63          | 28.15   |
> | CrossThink-QA | + RAVR (Ours)   | **25.76**          | **33.95**            | **20.42**            | 18.75            | **42.97**           | **31.89**            | **29.86**        | **28.51**          | **28.96**   |
>
>
> **Generalizability experiments on Qwen3-4B.**
>
>
> | Training Set | Model           | GPQA-D (Avg@4) | MMLU-Pro (Avg@4) | AIME 24 (Avg@16) | AIME 25 (Avg@16) | AMC 23 (Avg@16) | Minerva (Avg@16) | General Task | Math Reasoning | Average |
> |-------------|-----------------|----------------|------------------|------------------|------------------|-----------------|------------------|--------------|----------------|---------|
> | DeepMath    | + GRPO          | 50.13          | 67.45            | 49.79            | 38.33            | 86.25           | 68.75            | 58.79        | 60.78          | 60.12   |
> | DeepMath    | + DAPO          | 50.38          | **68.40**            | 47.29            | 37.71            | 85.63           | 67.64            | 59.39        | 59.57          | 59.51   |
> | DeepMath    | + RAVR (Ours)   | **52.65**          | 67.88            | **50.21**            | **39.79**            | **87.19**           | **69.12**            | **60.27**        | **61.58**          | **61.14**   |
>
>
>
> **Details.**
>
> * **More diverse benchmarks.**
>   We now evaluate RAVR on StrategyQA, TheoremQA, WebInstruct, Olympiad-Math, and Olympiad-Physics (Table 2). These benchmarks require compositional and theorem-driven reasoning rather than simple multiple-choice pattern matching. When trained on both CrossThink-QA and DeepMath, RAVR improves the average accuracy over GRPO in almost all settings (e.g., +0.67 and +1.22 avg@4 points on the two training regimes).
>
> * **Additional model families.**
>   We further train **DeepSeek-R1-Distill-1.5B** on CrossThink-QA and compare RAVR with GRPO (Table 3). RAVR outperforms GRPO on 5/6 evaluation datasets and improves the overall average from 28.15 to 28.96 (≈ +2.9% relative).
>
> * **Larger and stronger model.**
>   We also test **Qwen3-4B** on the more challenging DeepMath training set, including both GRPO and DAPO as baselines (Table 4). RAVR achieves the best average performance (61.14) and improves over GRPO and DAPO by about +1.0 and +1.6 absolute points (≈ +1.7% and +2.7% relative), respectively, while maintaining strong generalization to out-of-domain benchmarks.

---

> > ### Author Response · Authors · 2025-11-27
> > **Rebuttal by Author (2/3)**
> >
> > ## 2. What is the verifiable signal of the reasoning path (z), and how does our method perform inference?
> >
> > **TL;DR.**
> > Yes, in our variational training phase, the input is the question (x) together with the reference answer (y*), and the model outputs a reasoning path (z). For each sampled z, we define its reward using a **reference-answer probability**: we feed the model a synthesized sequence such as
> > “`[system prompt][user prompt with x] <think> z </think> The final answer is y*`”
> > and take the token log-probabilities of (y*) as the reward of that reasoning path. Intuitively, if a reasoning path (z) is sound and aligned with the problem, it should make the model more confident in the correct answer, hence a higher reward. We then combine this reward with GRPO-style group normalization (sampling multiple reasoning paths (z) for the same prompt and computing advantages within the group) to train the LLM to generate better reasoning when conditioned on both (x) and (y*), i.e., to improve LLM(z|x, y*).
> > In parallel, we add a KL divergence term between the **answer-conditioned** distribution LLM(z|x, y*) and **question-only** reasoning distribution LLM(z|x). This KL term is not ad-hoc: it arises directly from our **variational inference objective**, where LLM(z|x, y*) serves as a variational surrogate for LLM(z|x). Optimizing this objective pulls the question-only distribution toward the answer-conditioned one, transferring the ability to produce better reasoning back to LLM(z|x). As a result, **at test time** the model behaves like a standard “thinking-mode” LLM: it only receives the question (x), samples a reasoning path from LLM(z|x), and then generates the final response—no reference answer is used at inference.
> >
> > **Details.**
> >
> >   Concretely, for each sample we:
> >
> >   1. Show the model the question (x) and the reference answer (y*), and ask it to reconstruct a plausible solution, producing a reasoning path (z), i.e., `[system prompt][user prompt with x and y]`, . The model will generate such a trajectory: `<think> leveraging both the question and the reference answer here, thinking why the reference answer is logically reasonable for the given question </think> output the first-person think-aloud monologue as if solving the given question from scratch here, which is the expected reasoning (z) when only the question is provided.`"
> >   2. Then, we extract the generated reasoning path z, i.e., the content after \</think\> in the above trajectory together with the question (x) and the reference answer y* to synthesize a new trajectory: `[system prompt][user prompt with x] <think> z </think> The final answer is y*`
> >   3. We look at the log-probability the model assigns to the correct answer (y*) in this trajectory. This value is used as the reward for (z).
> >  4. For each question, we sample a **group** of reasoning paths (e.g., 8 different (z)’s) and then apply GRPO’s group-based normalization: reasoning paths that help much more than their peers get positive advantages, those that help less get negative advantages. This drives the model to generate more of the genuinely helpful reasoning and fewer unhelpful or spurious paths.
> >  5. We pull the reasoning token probability in steps 1 and 2 closer, which is the KL Divergence objective introduced by the variational inference. You can think of this as a “teacher–student” relationship rooted in variational inference:
> >
> >   * The *teacher mode* (seeing both question and answer) learns to produce very helpful reasoning because it is directly optimized to maximize the answer probability.
> >   * The *student mode* (seeing only the question) is trained to stay close to the teacher through the KL term that naturally appears in the variational objective.
> >
> >   After training, the student model alone is strong enough: during evaluation, we only use the question-only prompt, let the model “think” by sampling from LLM(z|x), and then produce the final answer. No ground-truth answer or posterior sampling is involved at inference, so from the user’s perspective, the model behaves like a normal thinking-mode LLM.

---

> > > ### Author Response · Authors · 2025-11-27
> > > **Rebuttal by Author (3/3)**
> > >
> > > ## 3. Applicability to tasks with ambiguous answers.
> > >
> > > **TL;DR.**
> > > Conceptually, *RAVR does not** require the answer to be unique: our analysis only assumes that the reference answer is logically correlated with the question and the reasoning path. This property makes RAVR particularly attractive for ambiguous, open-ended tasks where there may not be a single “correct” answer (e.g., writing or open-ended generation). In such settings, existing RL pipelines typically rely on a separate reward model (e.g., a large LLM judge or a reward model trained from human feedback), which incurs substantial additional cost and is susceptible to bias and reward hacking (see, e.g., *Justice or Prejudice? Quantifying Biases in LLM-as-a-Judge*, ICLR 2025). By contrast, RAVR does **not** require any external reward model: the reward for reasoning is defined via the model’s own likelihood of the reference answer given the reasoning, which naturally extends to multiple or “soft” reference answers. Concurrent work such as *Reverse-Engineered Reasoning for Open-Ended Generation* has already shown that leveraging reference-answer probabilities can be effective for reasoning quality assessment in writing tasks, which further supports the applicability of this family of methods beyond strictly verifiable tasks.
> > >
> > >
> > > **Details.**
> > >
> > > **Why Our Benchmark Choice is Reasonable.**
> > >   Our experimental setup follows the prevalent practice in *reinforcement learning for LLM reasoning*, where benchmarks are chosen to have clear, verifiable answers so that automatic evaluation is reliable and comparable across methods. For example, *General-Reasoner: Advancing LLM Reasoning Across All Domains* (NeurIPS 2025) evaluates on MMLU-Pro, SuperGPQA, GPQA, TheoremQA, BBEH, MATH500, Olympiad, Minerva, GSM8K, AMC23, AIME24, and AIME25—**all** of which are tasks with clearly defined reference answers (and most of them have been adopted by our work), precisely because they are well-suited to “Reinforcement Learning with Verifiable Reward (RLVR)”–style practices such as the DeepSeek-R1-Zero. **Our work is fully aligned with this protocol and, in fact, uses 11 datasets**, making our empirical coverage comparable to or broader than recent RL4LLM work. We agree that extending RAVR to fully open-ended benchmarks would be valuable, and we now explicitly state this as an important direction for future work, but we also note that our current evaluation setting is consistent with standard RL-for-reasoning practice.

---

### Official Review · Reviewer_n2it · 2025-10-31

**Soundness:** 3
**Presentation:** 2
**Contribution:** 3
**Rating:** 4
**Confidence:** 4

**Summary:**

This paper proposes RAVR (Reference-Answer-guided Variational Reasoning), an end-to-end training framework that leverages the reference answer during training to improve LLM's question-only reasoning at inference. The core idea is to treat answer-conditioned reasoning as a variational surrogate for the question-only prior.

The authors first formalize why conditioning on the answer should bias sampling toward higher-utility reasoning paths: using Bayes rule, the posterior over reasoning traces reweights the prior, leading to a provable increase in both the probability of above-average traces and the expected utility under the posterior. They then derive an ELBO target that maximizes answer likelihood under the posterior while minimizing a KL from posterior to prior, and instantiate a practical objective with (i) a utility-baseline reward measuring posterior improvement over the prior, (ii) KL sample reweighting by the utility, and (iii) a simple answer-prefix (“The answer is …”) trick for stable probability estimation.

Prompt templates are provided to make posterior traces read like first-person, think-aloud reasoning without revealing access to the answer. On Qwen3-1.7B, RAVR shows consistent gains across GPQA-Diamond, MMLU-Pro, AIME24/25, AMC23, and Minerva, and appears more sampling-efficient than GRPO (similar accuracy with 8 vs 24 rollouts).

**Strengths:**

1. Framing answer-conditioned reasoning as an amortized posterior that teaches the question-only prior via an ELBO-style objective, plus practical touches (utility-baseline, KL reweighting, answer-prefix, posterior “think-aloud” instructions), is novel. I mean using answers to induce better reasoning is standard, but incorporating it into training objectives is interesting.
2. The theoretical part is clearly derived， posterior re-weighting by s(z) implies a larger mass on good traces and a higher expected utility.
3. Empirically, RAVR improves Qwen3-1.7B's average over RL baselines (GRPO, DAPO, VeriFree, RLPR) across both general and math setups.

**Weaknesses:**

1. I am still concerned that models can inflate likelihood via stylistic cues ("answer-prefix"), instead of improving true correctness. The paper partly addresses this with a baseline and length normalization, but there is no more discussion.
2. Not sure if RAVR's advantage persists under equal wall-clock or token budgets as RL baselines.
3. All results use Qwen3-1.7B. If during rebuttal the authors can show similar improvements on other and even better larger models, I will raise my score.
4. Numerous minor grammar/typo issues ("refernce answe", "postive", "noval").

**Questions:**

See above.

---

> ### Author Response · Authors · 2025-11-24
> **Rebuttal by Author (1/3)**
>
> We appreciate your positive assessment of our work’s novelty, the clarity of the theoretical derivation, and the empirical gains. Below, we first summarize your main concerns and then respond to each, with a brief **TL;DR** followed by a more detailed discussion.
>
> ---
>
> ## 1. The role of the answer-prefix
>
> **TL;DR.**
> In our implementation, the answer-prefix is a *fixed, forced* sequence used solely to place all reasoning paths into the **same standardized answer-ready state** when computing reference-answer probability rewards during training. By design, this prefix is not intended to directly improve correctness, nor does it create opportunities to artificially inflate likelihood via stylistic cues. Instead, it enables a fair comparison of rewards across diverse reasoning paths, which is crucial for group-relative style reward normalization. As shown in Figure 6, our ablation **without** the answer-prefix achieves **comparable peak performance** but exhibits **substantially more unstable training curves**, suggesting that the prefix stabilizes reward evaluation.
>
> **Details.**
>
> * In many prior methods, during training, the reference-answer probability reward is computed by directly appending the reference answer y* after the reasoning path z, i.e., scoring z based on the token probabilities of y* in the sequence  "... \<think\> z \</think\> y*". The transition from the end of $z$ to the beginning of the first token after \</think\> is unconstrained and can vary across traces, which introduces bias due to differences in their tendencies for the first token generation after \</think\>.
>
> * In RAVR, we intentionally **standardize this transition** by simply inserting a fixed cue phrase. The sequence becomes
>   "... \<think\> z \</think\> The answer is y*". Because this prefix is **identical** for all reasoning paths, we compare conditional probabilities from a **uniform starting state**. The reward, therefore, depends only on how the reasoning path z helps the model assign a higher probability to the reference answer from this shared state. At **inference time**, the prefix is not used at all: models are evaluated by prompting the question and decoding the answer in the usual way.
>
> * In the ablation study (Figure 6), we trained a variant that **removes** the answer-prefix and computes the reference answer probability exactly as in prior work, i.e., on the sequence "... \<think\> z \</think\> y*". We observe that although the best checkpoint of this ablation can approach the full RAVR performance (their peak accuracies are similar), the evaluation accuracy curves fluctuate noticeably more across training steps, making the method more sensitive to early stopping and to checkpoint selection.

---

> ### Author Response · Authors · 2025-11-24
> **Rebuttal by Author (2/3)**
>
> ## 2. Fairness under equal token budgets
>
> **TL;DR.**
> We now report average token usage in the main results in Table 1 and observe that **RAVR attains better or comparable accuracy with similar or *lower* reasoning-token budgets than RL baselines**. And we added a discussion of the token budgets in Section 3.1
>
> **Details.**
>
> * We have updated **Table 1** to include, for each dataset and method, the **average token length** as a subscript to the accuracy. This makes the token budget directly visible. Part of the results comparing the on-par performance baselines with our method are shown below:
>
> | Training Set      | Model           | GPQA-D (Avg@4)   | MMLU-Pro (Avg@4) | AIME 24 (Avg@16) | AIME 25 (Avg@16) | AMC 23 (Avg@16)     | Minerva (Avg@16)    |
> |-------------------|-----------------|------------------|------------------|------------------|------------------|----------------------|---------------------|
> | **CrossThink-QA** | + GRPO          | 34.97 (4581)     | 55.18 (2587)     | 25.63 (7273)     | 23.54 (7283)     | **64.53 (4799)**    | 56.92 (3231)        |
> |                   | + DAPO          | 35.35 (5991)     | 54.75 (5129)     | 23.33 (7556)     | 21.88 (7495)     | 62.50 (5502)        | 56.80 (4129)        |
> |                   | + RAVR (Ours) | **40.91 (4177)** | **55.88 (2430)** | **27.92 (7236)** | **23.75 (7281)** | 62.03 (5073)        | **58.00 (3320)**    |
> | **DeepMath**      | + GRPO          | 33.21 (3932)     | 54.55 (1934)     | 26.67 (6935)     | 21.46 (6906)     | 67.03 (4602)        | 56.99 (2766)        |
> |                   | + DAPO          | **34.84 (5242)** | 55.25 (2684)     | 26.67 (7263)     | 23.33 (7070)     | 67.50 (5043)        | 57.97 (3633)        |
> |                   | + RAVR (Ours) | 34.60 (4566)     | **55.50 (2538)** | **29.17 (6593)** | **22.71 (6834)** | **69.69 (4615)**    | **58.43 (3261)**    |
>
>
>
> * Based on this table, we added the following analysis:
>
>   > *We further examine how the average reasoning length relates to performance. Overall, all RL-based methods substantially improve accuracy over the vanilla model while typically using fewer tokens, and RAVR tends to be more token-efficient: when trained on CrossThink-QA, RAVR saves more tokens than GRPO on 4 out of 6 datasets; when trained on DeepMath, RAVR uses an on-par token budget with GRPO and fewer tokens than DAPO. These observations suggest a reasonable length regime where chains are detailed enough to support correct reasoning but not overly verbose. RAVR tends to stay in this regime, likely because answer-conditioned reasoning, guided by the reference answer, prunes redundant or overthinking segments and encourages compact yet effective reasoning during training.*

---

> ### Author Response · Authors · 2025-11-24
> **Rebuttal by Author (3/3)**
>
> ## 3. Generalization to Other and even Larger and Stronger Models
>
> **TL;DR.**
> We added new experiments in Section 3.2 on **DeepSeek-R1-Distill-1.5B** (a different model family) and **Qwen3-4B** (a stronger and larger model). In both cases, RAVR consistently outperforms strong RL baselines: roughly **+2.9%** over GRPO on DeepSeek-R1-Distill-1.5B, and **+1.7% / +2.7%** over GRPO / DAPO on Qwen3-4B. These results support the claim that the proposed method generalizes across both architectures and scales.
>
> **Details.**
>
> To address the request for broader validation, we conducted additional experiments summarized in a new Section 3.2, *“Generalizability Analysis of RAVR”*.
>
> * **DeepSeek-R1-Distill-1.5B**
>   Using CrossThink-QA as the training set, we compared RAVR with GRPO (the strongest baseline in previous results):
>
>   | **Training Set**  | **Model**     | **GPQA-D (Avg@4)** | **MMLU-Pro (Avg@4)** | **AIME 24 (Avg@16)** | **AIME 25 (Avg@16)** | **AMC 23 (Avg@16)** | **Minerva (Avg@16)** | **General Task** | **Math Reasoning** | **Average** |
>   | ----------------- | ------------- | ------------------ | -------------------- | -------------------- | -------------------- | ------------------- | -------------------- | ---------------- | ------------------ | ----------- |
>   | **CrossThink-QA** | + GRPO        | 24.75              | 33.65                | 18.75                | 19.17                | 41.88               | 30.70                | 29.20            | 27.63              | 28.15       |
>   |                   | + RAVR (Ours) | **25.76**          | **33.95**            | **20.42**            | 18.75                | **42.97**           | **31.89**            | **29.86**        | **28.51**          | **28.96**   |
>
>   > *Overall, RAVR demonstrates consistent improvements. On DeepSeek-R1-Distill-1.5B, RAVR outperforms GRPO on 5 out of 6 datasets, yielding an overall improvement of approximately 2.9%.*
>
> * **Qwen3-4B**
>   Qwen3-4B is substantially stronger; its accuracy on CrossThink-QA is close to 90%, leaving limited headroom. To stress-test the methods, we therefore used the more challenging **DeepMath** training set and included **DAPO** (which is particularly competitive on math) as an additional baseline:
>
>   | **Training Set** | **Model**     | **GPQA-D (Avg@4)** | **MMLU-Pro (Avg@4)** | **AIME 24 (Avg@16)** | **AIME 25 (Avg@16)** | **AMC 23 (Avg@16)** | **Minerva (Avg@16)** | **General Task** | **Math Reasoning** | **Average** |
>   | ---------------- | ------------- | ------------------ | -------------------- | -------------------- | -------------------- | ------------------- | -------------------- | ---------------- | ------------------ | ----------- |
>   | **DeepMath**     | + GRPO        | 50.13              | 67.45                | 49.79                | 38.33                | 86.25               | 68.75                | 58.79            | 60.78              | 60.12       |
>   |                  | + DAPO        | 50.38              | **68.40**            | 47.29                | 37.71                | 85.63               | 67.64                | 59.39            | 59.57              | 59.51       |
>   |                  | + RAVR (Ours) | **52.65**          | 67.88                | **50.21**            | **39.79**            | **87.19**           | **69.12**            | **60.27**        | **61.58**          | **61.14**   |
>
>   > *On Qwen3-4B with DeepMath, the proposed method consistently surpasses GRPO and outperforms DAPO on 5 out of 6 datasets, with overall improvements of about 1.7% over GRPO and 2.7% over DAPO.*
>
> These results indicate that the core idea—using answer-conditioned posteriors to teach a question-only prior—remains effective across **different model families** and **larger model scales**.
>
> ## 4. Minor grammar and typo issues
>
> **TL;DR.**
> We have carefully proofread the manuscript and corrected all identified typos and minor grammar issues.
>
> **Details.**
>
> We fixed the specific errors you pointed out and performed an additional language pass to correct similar issues. The revised draft should read more smoothly without affecting any technical content.

---

### Official Review · Reviewer_5oq8 · 2025-11-01

**Soundness:** 2
**Presentation:** 2
**Contribution:** 2
**Rating:** 4
**Confidence:** 2

**Summary:**

The paper proposes RAVR, a reference-answer-guided RL framework that enhances LLM reasoning on math and logic benchmarks. RAVR trains the model using an answer-conditioned “posterior” distribution that maximizes the likelihood of the reference answer while remaining close to the question-only “prior,” incorporating a prior-based utility baseline, reward-weighted KL, a think-aloud prompt, and an answer-prefix cue, end-to-end optimized via GRPO. On Qwen3-1.7B, RAVR achieves notable gains over previous baselines: when trained on CrossThink-QA it reaches 40.91 GPQA-Diamond (+5.56 over DAPO) with the best overall average of 44.75, and when trained on DeepMath it obtains the top overall score of 45.02, demonstrating strong out-of-domain transfer.

**Strengths:**

+ The paper introduces an answer-conditioned variational objective that meaningfully improves exploration in RL-based reasoning, offering a conceptual advance over prior reward-only or STaR-style methods.

+ The experiment results show consistent performance gains across multiple reasoning benchmarks, supported by various ablations and behavioral analyses that validate the source of improvement.

**Weaknesses:**

+ RAVR treats the reference-answer token probability as the reward signal for the generated reasoning trace. However, this self-supervised likelihood-based reward is known to be noisy and susceptible to reward hacking, where models can optimize for probability shortcuts and could easily lead to training collapse.


+ The method depends on having gold reference answers during training, which may limit its applicability in weakly supervised or open-ended settings where answers are unavailable or ambiguous.


+ While the paper reports gains on the selected benchmarks, it does not provide qualitative evaluations to show whether the performance transfers beyond multiple-choice and math tasks to other broader settings that require strong reasoning capabilities.

**Questions:**

See above.

---

> ### Author Response · Authors · 2025-11-27
> **Rebuttal by Author (1/3)**
>
> Thank you for recognizing both the conceptual contribution of the answer-conditioned variational objective and the consistent empirical gains supported by ablations and behavioral analyses. Below, we first summarize your main concerns and then respond to each, with a brief TL;DR followed by a more detailed discussion.
>
> ---
> ## 1. Reliability of the likelihood-based reward
>
> **TL;DR.**
> Based on our observations, the likelihood-based reward in RAVR is well-behaved and does not cause reward hacking or collapse:
>
> - We conducted a new analysis on hard CrossThink-QA questions where the base model fails under question-only reasoning. On these questions, **logically valid (i.e., free of hallucinations or reward hacking) and independent (i.e., not indicating prior access to the reference answer) reasoning paths yield the largest gains in answer log-probability**, and significantly outperform both invalid types (p < 0.05), even though many invalid paths also end with the correct answer.
> - During training, the reward and held-out validation accuracy curves increase smoothly (Figures 4 and 5), without the “reward explosion + accuracy collapse” pattern typically associated with reward hacking and training collapse. **Our ablation study suggests that these gains mainly stem from the design of the reward baseline, the posterior instruction, and the answer prefix.**
> - RAVR’s **reasoning lengths remain moderate and roughly aligned with task difficulty** (Table 1), ruling out the simple strategy of “just make chains arbitrarily long or find shortcuts” to game the reward.
> - A small penalty on explicit “reference-answer leakage” further suppresses degenerate trajectories.
>
> Together, these observations show that the likelihood-based reward in RAVR is a stable and informative supervision signal.
>
> ---
>
> **Details**
> We added a new section, *Rationality Analysis of Answer-Conditioned Reasoning*, in the appendix. We first identify 85 challenging questions from 500 questions in CrossThink-QA. On these questions, the model generates answer-conditioned reasoning paths, which are categorized into four types (by GPT-5.1) based on whether the reasoning is logically valid (i.e., free of hallucinations or reward hacking) and whether it is independent of the reference answer (i.e., not indicating prior access). For each example and type, we compute the average improvement in log-probability of the ground-truth answer over the question-only baseline.
>
> > **Table 1.** Δ log p for each answer-conditioned reasoning type.
>
> | Reasoning type            | n  | mean | std  | min   | 25%   | 50%  | 75%  | max   |
> |---------------------------|----|------|------|-------|-------|------|------|-------|
> | Valid & Independent       | 53 | 4.97 | 4.25 | -7.18 |  3.63 | 5.45 | 7.41 | 12.38 |
> | Valid & Dependent         | 31 | 4.11 | 4.36 | -6.23 |  0.54 | 5.06 | 7.29 | 11.03 |
> | Invalid & Independent     | 62 | 2.09 | 3.93 | -7.71 | -0.72 | 1.20 | 5.16 | 10.80 |
> | Invalid & Dependent       | 59 | 3.27 | 4.15 | -6.38 | -0.04 | 3.97 | 5.95 | 13.65 |
>
> **Valid & independent reasoning achieves the highest mean and median Δ log p and the strongest lower tail (25% 3.63)**, indicating both strong and stable gains on these hard questions.
>
> To compare types on the same examples, we run paired t-tests between valid & independent reasoning and each of the other three types:
>
> > **Table 2.** Paired t-tests on per-example Δ log p.
>
> | Comparison                                     | n  | t     | p      |
> |-----------------------------------------------|----|-------|--------|
> | Valid & Independent vs. Valid & Dependent     | 28 | -1.54 | 0.134  |
> | Valid & Independent vs. Invalid & Independent | 34 |  2.93 | 0.006  |
> | Valid & Independent vs. Invalid & Dependent   | 32 |  2.15 | 0.039  |
>
> Valid & independent reasoning **significantly outperforms both invalid types (p < 0.05)**, confirming that the reward systematically favors logically sound reasoning rather than hallucinated or reward-hacked trajectories, even when the latter end with the correct answer. For the valid & dependent ones, which are most likely to introduce negative side effects, our qualitative inspection reveals a typical failure mode: the reasoning explicitly refers to the provided answer, using phrases such as “according to the reference answer”. To discourage such behavior, we introduce an additional reward term that penalizes explicit leakage: if a trajectory contains the substring “reference answer”, we assign an extra penalty of $-0.5$ to its reward. This effectively separates valid & independent reasoning from valid & dependent ones. The evolution of reference-answer leakage during training is shown in Figure 7: as training progresses, the proportion of reasoning paths that explicitly mention the reference answer steadily decreases, and at inference time, when the model does not receive the reference answer, we do not observe such leakage in the generated reasoning.

---

> ### Author Response · Authors · 2025-11-27
> **Rebuttal by Author (2/3)**
>
> ## 2. experiments on more benchmarks.
>
> **TL;DR.**
> We added a new section **3.2 Generalizability Analysis of RAVR** with *five more diverse benchmarks* beyond standard multiple-choice/math tasks. RAVR consistently improves over strong RL baselines, suggesting that the method generalizes across datasets and domains.
>
>
> **Generalizability experiments on more benchmarks.**
>
> | Training Set  | Model           | StrategyQA (Avg@4) | TheoremQA (Avg@4) | WebInstruct (Avg@4) | Olympiad-Math (Avg@4) | Olympiad-Physics (Avg@4) | Average |
> |--------------|-----------------|--------------------|--------------------|----------------------|------------------------|---------------------------|---------|
> | CrossThink-QA | + GRPO          | 64.25              | 58.04              | 75.52                | 56.23                  | **13.14**                     | 53.44   |
> |  | + RAVR (Ours)  | **66.17**              | **58.50**              | **76.27**                | **56.67**                  | 12.95                     | **54.11**   |
> | DeepMath      | + GRPO          | 62.75              | 57.33              | 72.67                | 58.17                  | 11.97                     | 52.58   |
> |      | + RAVR (Ours)   | **63.33**              | **57.73**              | **75.83**                | **59.83**                  | **12.29**                     | **53.80**   |
>
> **Details.**
>
> * **More diverse benchmarks.**
>   We now evaluate RAVR on StrategyQA, TheoremQA, WebInstruct, Olympiad-Math, and Olympiad-Physics (Table 2). These benchmarks require compositional and reasoning rather than simple multiple-choice and math tasks. For example, TheoremQA is a theorem-driven QA benchmark that tests the ability to apply formal theorems to solve challenging science problems.  When trained on both CrossThink-QA and DeepMath, RAVR improves the average accuracy over GRPO in almost all settings (e.g., +0.67 and +1.22 avg@4 points on the two training regimes).
>
> * **Why Our Benchmark Choice is Reasonable.** Our experimental setup follows the prevalent practice in *reinforcement learning for LLM reasoning*, where benchmarks are chosen to have clear, verifiable answers so that automatic evaluation is reliable and comparable across methods. For example, *General-Reasoner: Advancing LLM Reasoning Across All Domains* (NeurIPS 2025) evaluates on MMLU-Pro, SuperGPQA, GPQA, TheoremQA, BBEH, MATH500, Olympiad, Minerva, GSM8K, AMC23, AIME24, and AIME25—**all** of which are tasks with clearly defined reference answers (and most of them have been adopted by our work), precisely because they are well-suited to “Reinforcement Learning with Verifiable Reward (RLVR)”–style practices such as the DeepSeek-R1-Zero. **Our work is fully aligned with this protocol and, in fact, uses 11 datasets**, making our empirical coverage comparable to or broader than recent RL4LLM work. We agree that extending RAVR to fully open-ended benchmarks would be valuable, and we now explicitly state this as an important direction for future work, but we also note that our current evaluation setting is consistent with standard RL-for-reasoning practice.

---

> ### Author Response · Authors · 2025-11-27
> **Rebuttal by Author (3/3)**
>
> ## 3. Dependence on Reference Answers and Applicability Beyond Fully Supervised Settings
>
> **TL;DR.**
> Using gold reference answers is one of the most **common practices** in RL-for-LLM research: prior works such as GRPO, DAPO, VeriFree, and RLPR all assume access to correct answers for reward computation or verification. RAVR operates under the *same* supervision regime. Moreover, RAVR does **not** require a single, deterministic answer: the reference signal can be a set of candidates or any informative text related to the preferred response, making the framework conceptually suitable for ambiguous or open-ended tasks.
>
> ---
>
> **Details.**
>
> **Reference answers as the standard setup.**
> Most recent RL-for-reasoning methods assume access to reference answers during training:
>
> - **GRPO / DAPO**: compute rewards by checking whether the final answer matches the gold label using a rule-based verifier.
> - **RLT (NeurIPS 2025), NOVER (EMNLP 2025), VeriFree, RLPR**: remove explicit verifiers but still use the **LLM’s probability of the reference answer** as the reward signal.
> - Even some *generative* reward models take the reference answer as input, e.g., **Polar: Pre-Trained Policy Discriminators are General Reward Models (NeurIPS 2025)**.
>
> RAVR follows the same supervision assumption as these methods; our contribution lies in **how** we use the reference answer: instead of merely treating it as a target for reward computation, we leverage it via an answer-conditioned variational objective to directly guide the learning of the reasoning distribution.
>
> **Ambiguous and open-ended tasks.**
> Conceptually, RAVR does **not** require the answer to be unique: our analysis only assumes that the reference answer is logically correlated with the question and the reasoning path. This makes RAVR particularly attractive for ambiguous, open-ended tasks where there may not be a single “correct” answer (e.g., writing or long-form generation).
>
> In such settings, existing RL pipelines typically rely on a separate reward model (e.g., a large LLM judge or a reward model trained from human feedback), which incurs substantial additional cost and is susceptible to bias and reward hacking (see, e.g., *Justice or Prejudice? Quantifying Biases in LLM-as-a-Judge*, ICLR 2025). By contrast, RAVR does not require any external reward model: the reward for reasoning is defined via the model’s own likelihood of the reference answer given the reasoning, which naturally extends to open-ended reference answers.
>
> Concurrent work such as *Reverse-Engineered Reasoning for Open-Ended Generation* has already shown that leveraging reference-answer probabilities can be effective for assessing reasoning quality in writing tasks, which further supports the applicability of this family of methods beyond strictly verifiable, single-answer benchmarks.

---

### Author Response · Authors · 2025-12-01
**General Response**

This work **addresses the exploration challenge in RL for LLM reasoning**, where the model fails to sample high-quality reasoning from the question alone and thus easily gets stuck in local optima. To this end, we introduce **RAVR, the first end-to-end framework that uses answer-conditioned reasoning as a variational surrogate for question-only reasoning**. We demonstrate the effectiveness of RAVR through extensive experiments on both general and math reasoning benchmarks, as well as comprehensive ablations and behavioral analyses that validate the source of improvement.

Overall, the reviewers found our work to be **well-motivated and novel**, **theoretically sound**, and supported by **logically structured empirical evaluations**, indicating that RAVR provides a principled and practically useful contribution to the RL-for-LLM reasoning community:

* **Motivation & novelty (5oq8, n2it, and 1Zeu).**
5oq8 notes that our objective *“meaningfully”* improves exploration and is a *“conceptual advance”* over prior methods; n2it describes our framing as *“novel”* and *“interesting”*; 1Zeu calls the integration *“innovative”*.

* **Theoretical soundness (n2it and 1Zeu).**
n2it states that *“the theoretical part is clearly derived”*, while 1Zeu emphasizes a *“solid theoretical foundation”*.

* **Experimental design (5oq8 and n2it).**
5oq8 highlights *“consistent performance gains across multiple reasoning benchmarks”* together with ablations and behavioral analyses that *“validate the source of improvement”*, and n2it points out that RAVR improves over *“strong RL baselines across both general and math setups”*.

The major concerns focus on **effectiveness across more benchmarks and models** and **applicability beyond tasks with explicit reference answers**. Our responses are summarized as follows:

* **Concern 1: Effectiveness across more benchmarks and models.**
Reviewers (5oq8, n2it, 1Zeu) requested additional experiments to more thoroughly test whether our gains hold across different evaluation benchmarks and models. Reviewer n2it even explicitly wrote that *“If during rebuttal the authors can show similar improvements on other and even better larger models, I will raise my score.”*

  **Our Response.** In the rebuttal, we added the following experiments and found that RAVR consistently outperforms strong RL baselines:

  1. **More benchmarks.** We evaluate on **five additional reasoning benchmarks**, including datasets beyond multiple-choice (e.g., TheoremQA) and beyond purely math tasks (e.g., Olympiad-Physics).
  2. **More models.** We evaluate **two additional models** from other families and scales (DeepSeek-R1-1.5B and Qwen3-4B).

  We also organize these new results into a new subsection, **Section 3.2 “Generalizability Analysis of RAVR”**, to make the empirical picture clearer. Overall, our experiments now cover **11 datasets**, making our empirical coverage comparable to, and in some cases broader than, prior baselines and recent RL4LLM work.

* **Concern 2: Applicability beyond tasks with explicit reference answers.**
Reviewers (5oq8, 1Zeu) also raised concerns about how RAVR applies when reference answers are unavailable or more open-ended.

  **Our Response.** We first clarify that **having reference answers is one of the standard supervision setups** in RL-for-LLM research: even recent general reward-model works such as General-Reasoner (NeurIPS 2025) and POLAR (NeurIPS 2025) assume access to correct answers for reward computation or verification, and RAVR operates under the *same* supervision regime. **Conceptually, RAVR does not require a single unique answer**: our analysis only assumes that the reference signal is logically correlated with the question and the reasoning path. This makes the framework naturally compatible with settings where there may not be a single “correct” answer (e.g., writing or explanation-style tasks).

  In fact, a concurrent work (REverse-Engineered Reasoning for Open-Ended Generation), which focuses only on writing tasks, has shown that leveraging reference-answer probabilities can be effective for assessing reasoning quality, providing further evidence that this family of methods can extend beyond strictly verifiable, single-answer data.

As for other minor concerns that were raised by only a single reviewer, we have also taken concrete steps to address them. For example, regarding whether the likelihood-based reward might be susceptible to reward hacking or lead to training collapse (5oq8), we added additional in-depth quantitative and qualitative analyses in Appendix **“A.3 Rationality Analysis of Answer-Conditioned Reasoning”**. For issues such as typos (n2it) and clarification questions about specific parts (1Zeu), we have made the necessary corrections and provided clearer explanations.

Overall, during the discussion period, we believe we have carefully addressed the concerns of all reviewers and further improved the quality and clarity of the paper.

---

### Note · Authors · 2026-01-06

I have read and agree with the venue's withdrawal policy on behalf of myself and my co-authors.